# Evidence for igneous differentiation in Sudbury Igneous Complex and impact-driven evolution of terrestrial planet proto-crusts

Rais Latypov [1], Sofya Chistyakova[1], Richard Grieve [2] & Hannu Huhma[3]

Bolide impact is a ubiquitous geological process in the Solar System, which produced craters and basins filled with impact melt sheets on the terrestrial planets. However, it remains controversial whether these sheets were able to undergo large-scale igneous differentiation, or not. Here, we report on the discovery of large discrete bodies of melanorites that occur throughout almost the entire stratigraphy of the 1.85-billion-year-old Sudbury Igneous Complex (SIC) – the best exposed impact melt sheet on Earth – and use them to reaffirm that conspicuous norite-gabbro-granophyre stratigraphy of the SIC is produced by fractional crystallization of an originally homogeneous impact melt of granodioritic composition. This implies that more ancient and compositionally primitive Hadean impact melt sheets on the Earth and other terrestrial planets also underwent large-volume igneous differentiation. The near-surface differentiation of these giant impact melt sheets may therefore have contributed to the evolution and lithological diversity of the proto-crust on terrestrial planets.

[1] School of Geosciences, University of the Witwatersrand, Johannesburg, South Africa. [2] Centre for Planetary Science and Exploration, University of Western Ontario, London, Ontario, Canada. [3] Geological Survey of Finland, Espoo, Finland. Correspondence and requests for materials should be addressed to R.L. (email: rais.latypov@wits.ac.za)

Hypervelocity impacts affected the early crustal evolution of all the terrestrial planets, predominantly though the formation of large multi-ring craters and basins[1]. Such impacts resulted in melting of target rocks to form superheated impact melt sheets up to tens of km thick and thousands of km in diameter that were largely contained within the basins. There is currently major disagreement as to the initial nature and subsequent evolution of these voluminous impact melt sheets[2–19], especially those within the multi-ring basins on the Moon[10–12,14]. These are not, however, yet directly accessible and, therefore, the ground-truth data on the nature and differentiation of the impact melt sheets on the Moon and the other terrestrial planets is exclusively provided by terrestrial data[1]. The melt body used as the closest terrestrial analog of these voluminous impact melts sheets is the Sudbury Igneous Complex (SIC) in Canada—the largest, best exposed and accessible impact melt sheet on Earth[2–4] (Fig. 1). The SIC resulted from a large impact ~1.85 billion years ago[20], which produced a superheated (at least, 1700–2000 °C)[8,21,22] melt sheet of granodioritic composition up to 5 km thick[2,3], with an estimated original volume of >10$^4$ km$^3$. The SIC shows a remarkable magmatic stratigraphy, with a basal layer of mafic composition (norite) overlain by a felsic (granophyre) layer, with a transitional (quartz gabbro) phase (Fig. 1). This compositional stratigraphy has been traditionally interpreted as resulting from internal differentiation of an originally homogeneous impact melt sheet[2–6], but this hypothesis has been challenged.

It has been argued, however, that internal differentiation of granodioritic melt in the SIC either cannot be effective[15–17] or it is even totally impossible[18,19]. Alternative models attribute the stratigraphy of the SIC to stratification of the impact melt sheet into separate melt layers prior to onset of crystallization, with an upper felsic one crystallizing granophyre and the lower mafic one producing norite. These scenarios might involve two melts of different origin, one from the crust and another from the mantle[15,16]; alternatively impact-induced density stratification into a lower layer with mafic clasts and the upper layer with felsic clasts[17], and impact-induced melting that gave rise to heterogeneous emulsion of immiscible liquids which segregated into the upper felsic layer and a lower mafic layer[18,19,23]. Such models suggest that the impact of large bolides does not readily cause homogenization of melted targeted crustal rocks, and that superheated melt in impact-generated sheets does not undergo internal differentiation into compositionally stratified magmatic bodies. If correct, such hypotheses nullify likely contributions of impact cratering processes to the evolution and diversification of melt lithologies in the early crust on terrestrial planets[7–12,14,24–26].

Here we reassess models for the formation of the SIC following the discovery of discrete bodies (10 to 100s meters in size) of melanorites in the SIC. These bodies occur throughout much of the stratigraphy of the SIC and are, even locally, developed along its roof. Our field and geochemical observations indicate that the melanorites are likely fragments of a roof (mela)noritic sequence of the SIC which initially grew from the top of the melt sheet downwards but was later disrupted and collapsed as blocks onto the temporary floor. The existence of an original (mela)noritic sequence refutes the two-layer stratification models because they only allow granophyre to crystallize from the roof. The contemporaneous inward growth of similar (mela)noritic sequences from the base and roof of the SIC indicates that, in response to complete mixing of melted target rocks by a bolide impact, the

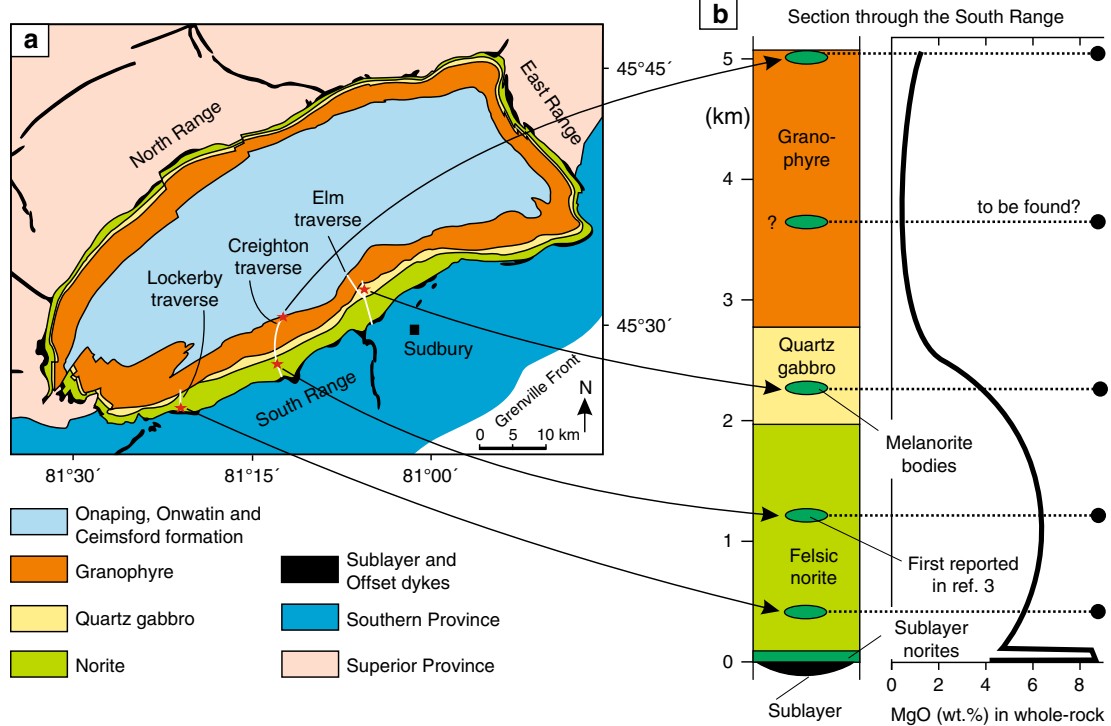

**Fig. 1** Schematic map and generalized section through the Sudbury Igneous Complex (SIC) showing the location of discovered melanorite bodies. **a** Schematic map of the SIC that is subdivided into two sectors referred to as the South Range, and the North and East Ranges. Map depicts the geometry of the main units and the location of individual sampling traverses (white lines) and discovered melanorite bodies (red stars). Map is simplified from ref. 2. **b** Generalized section through the SIC along the Creighton traverse of the South Range showing its stratigraphic subdivision, the position of melanorite bodies and whole-rock MgO geochemistry. The melanorite body first reported in ref. [3] is indicated. Note that in terms of whole-rock MgO all melanorite bodies are only comparable with the basal Sublayer norites. Section is simplified from ref. [3,35]

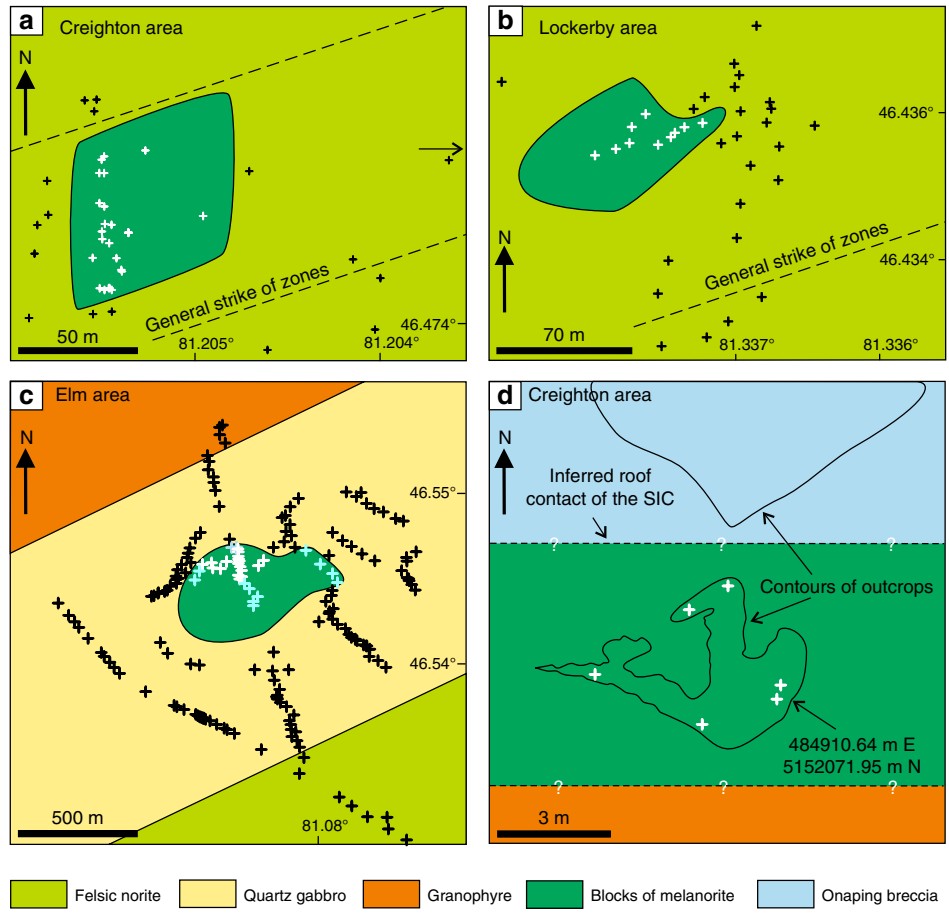

**Fig. 2** Schematic maps showing melanorite bodies in the South Range of the SIC. **a** A medium-sized melanorite body hosted by felsic norites in the Creighton area. **b** A medium-sized melanorite body hosted by felsic norites in the Lockerby area. **c** The largest melanorite body hosted by quartz gabbro in the Elm area. **d** Outcrop of the melanorite along the roof of the SIC in the Creighton area (mapped by Walter Peredery, 2017). Samples of the bodies with MgO = 7–9 wt.% are indicated by white crosses whereas those with MgO = 5-6 wt.% are indicated by blue crosses; black crosses indicate all host rocks of the SIC. The discrete shape of the bodies in **a–c** indicates that these are separate blocks rather than (semi)continuous layers in the stratigraphy of the SIC

Sudbury impact melt sheet was originally homogeneous. This implies that the spectacular compositional variations with stratigraphic height in the SIC are the result of fractional crystallization. An important implication is that more ancient and primitive Hadean impact melt sheets—comparable in size to the lunar ones[10–12,14]—on the early Earth and other terrestrial planets would also have undergone near-surface, large-volume differentiation to produce compositionally stratified bodies. Foundering of the lower ultramafic cumulates of these bodies may have contributed to the lithological diversification (e.g., formation of the considerable volumes of mafic-to-felsic rocks) during the early crustal evolution of the terrestrial planets[24–26], in addition to endogenic geologic processes involving recycling of primitive basaltic crust by deep geodynamic processes[27–32].

## Results

### A first melanorite body found in the SIC.

The South Range of the SIC (Fig. 1a) consists of five major rock units (from base to top): (1) Sublayer[33] representing a discontinuous group of magmatic breccias in depressions which contain magmatic Ni–Cu–PGE sulfide ores and connected in places with radial Offset dykes[34]; (2) Sublayer norites tending to only occur in the embayments[2] developed along the base and made up of cumulus orthopyroxene along with interstitial plagioclase, clinopyroxene and quartz; (3) felsic norite composed of cumulus orthopyroxene and plagioclase with some amount of interstitial clinopyroxene, (4) quartz gabbro characterized by disappearance of orthopyroxene and successive appearance of cumulus clinopyroxene, amphibole, magnetite, ilmenite and apatite; and (5) granophyre characterised by arrival of intergrowths of quartz and potassium feldspar[2,3,35] (Fig. 1b). The most recent and puzzling finding in this stratigraphy of the South Range is an abrupt reversal towards more MgO-rich rock compositions that manifests as a horizon of medium-grained, sulphide-bearing melanonorite ~1.3 km above the base of the SIC in the Creighton area[2,3] (Figs. 1, 2a and 3). The term melanorite was introduced to emphasize a higher amount of orthopyroxene (from 23 to 30–35 vol.%) compared to more normal felsic norite (from 6 to 8–9 wt.%)[2,3]. Unlike adjacent noritic rocks that are almost devoid of sulphides, melanorites commonly contains blebs of pyrrhotite-pentlandite-chalcopyrite sulphides (up to 1 wt.%; S = 0.15–0.30 wt.%) (Fig. 3).

The occurrence of the sulphide-bearing melanorite so high in the sequence of felsic norites is an anomaly, as rocks of such primitive composition with sulphides were believed to occur exclusively at the very base of the SIC, where they overly the Sublayer[2,3,33] (Fig. 1b). We have documented that this reversal along the Creighton traverse does not form a continuous layer in the magmatic stratigraphy of the SIC but is the result of a discrete body of melanorite (~60–80 m), which terminates along strike in both directions (Fig. 2a). Geochemical mapping of the same locality by other researchers yielded similar results[36]. No other

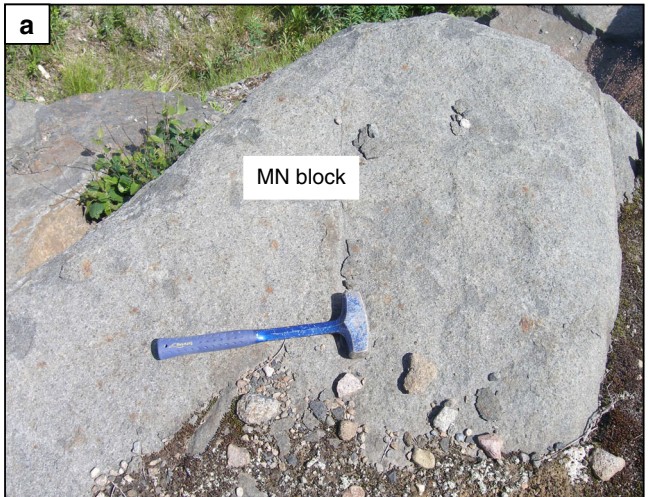

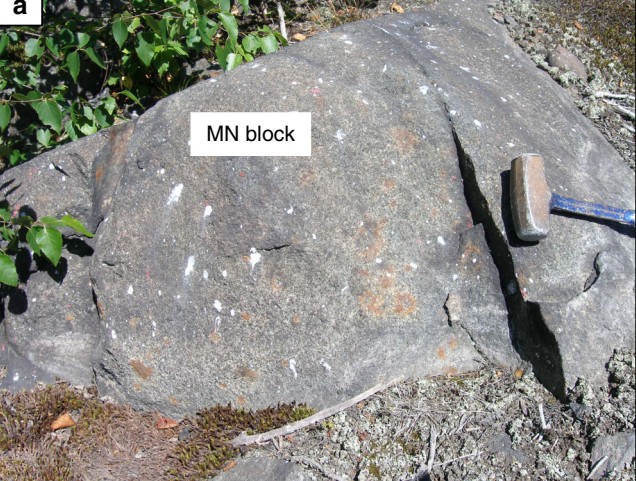

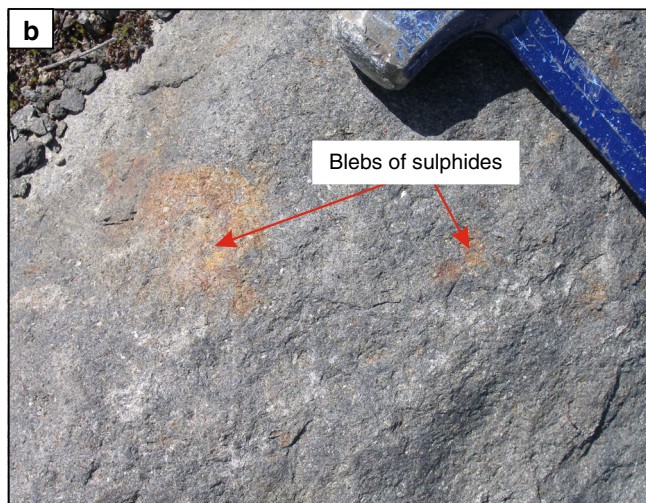

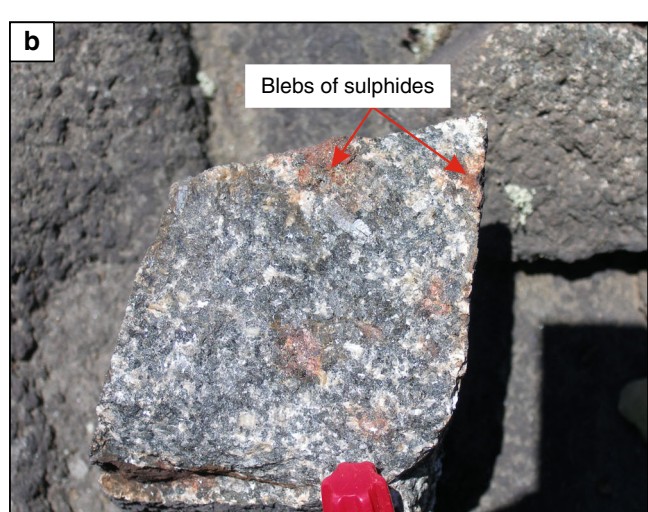

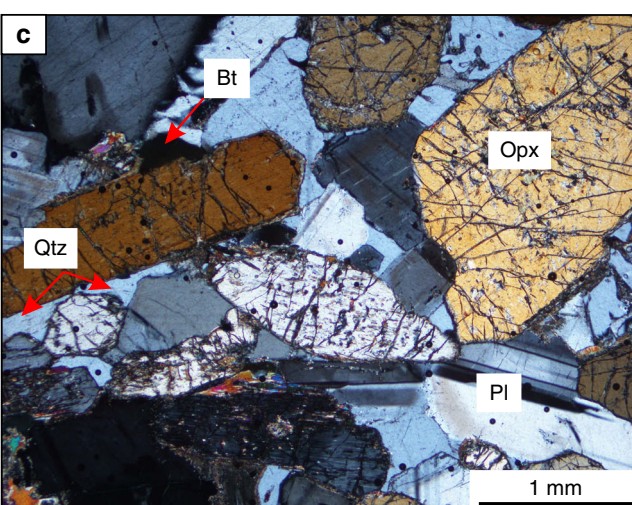

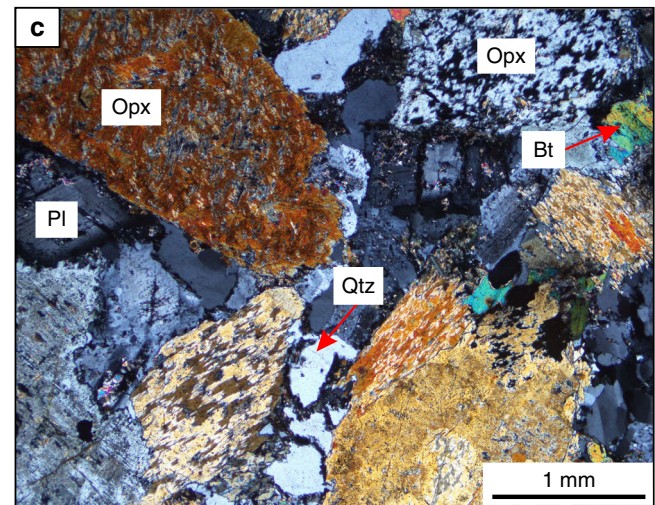

**Fig. 3** Melanorite body hosted by felsic norite of the Creighton traverse in the South Range of the SIC. **a** Field photograph of an outcrop of melanorite body hosted by felsic norite. The length of the hammer head used for scale is 11 cm. **b** Blebs of sulphides on a weathered surface of melanorite. **c** Photomicrograph showing euhedral/subhedral orthopyroxene and mostly subhedral plagioclase with minor interstitial biotite and quartz. Sample J-24 in cross-polarized light. Here and in text: Opx–orthopyroxene, Pl–plagioclase, Cpx–clinopyroxene, Qtz–quartz, Bt–biotite, Ksp–alkali feldspar. This body of melanorite was first reported in ref. [3] (Fig. 1b)

**Fig. 4** Melanorite body hosted by quartz gabbro of the Elm traverse in the South Range of the SIC. **a** Field photograph of an outcrop of sulphides-bearing melanorite body hosted by quartz gabbro. The length of the hammer head used for scale is 11 cm. **b** Blebs of sulphides on a fresh surface of melanorite. The length of a marker used for scale is 1 cm. **c** Photomicrograph showing extensive alteration of euhedral/subhedral orthopyroxene and subhedral plagioclase with minor interstitial biotite and quartz. Sample 480 in cross-polarized light

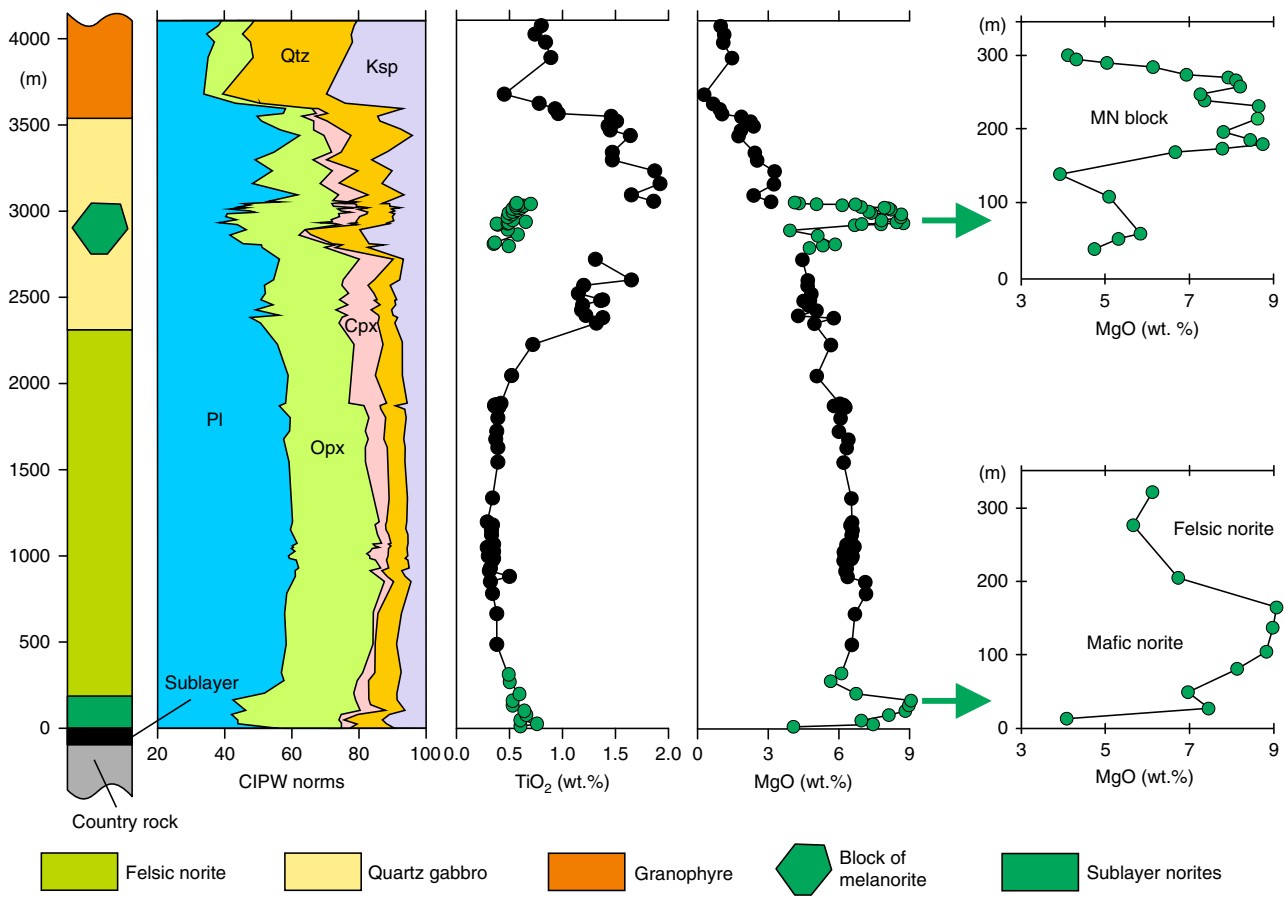

**Fig. 5** Position of the largest melanorite body in the chemical stratigraphy of the SIC. Simplified section through the SIC along the Elm traverse, South Range, consists of felsic norites, quartz gabbro and granophyre. The body is located within the quartz gabbro (see Figs. 1 and 2c) and is distinguished by lower contents of whole-rock TiO$_2$ and higher MgO contents compared to the host quartz gabbro. The body is compositionally more primitive than felsic norites of the SIC, except for mafic norites that overly the Sublayer. Note that the body shows a compositional zonation in terms of MgO that is mirrored by that of mafic and felsic norites overlying the Sublayer (all indicated by green symbols). Original data are located in Supplementary Data 1

melanorite bodies have been found within 1.5 km along strike from this melanorite body. It should be noted that visual identification of this melanorite body in the field is quite difficult. Even petrography is not entirely diagnostic. Geochemistry appears to be the best tool to allow the confident identification and mapping of the melanorite body. This explains why, despite an intensive study of the SIC for more than a century, this melanorite body has been overlooked.

**New melanorite bodies revealed by geochemical mapping**. We have undertaken several geochemical traverses and detailed mapping involving more than 600 whole-rock chemical analyses. This allowed us to document a number of new bodies of melanorites at other stratigraphic levels of the South Range. In particular, one such melanorite body (~50–60 m) occurs at ~0.7 km from the base of the SIC within the lower portion of norite at Lockerby (Figs. 1 and 2b). Another and, so far, the largest (~400–700 m) is located at ~2.7 km above the base within quartz gabbro in the Elm area (Figs. 1 and 2c). This melanorite (Fig. 4) is medium-grained rock with blebs of sulphides clearly visible on a weathered surface. Main minerals are euhedral/subhedral orthopyroxene (up to 36 vol. %) as prismatic crystals and the euhedral/subhedral plagioclase (up to 50 vol. %) as large, tabular grains. Both orthopyroxene and plagioclase are highly altered. This body is rich in MgO (up to 8–9 wt.%) and poor in TiO$_2$ (0.3–0.6 wt.%) and is, therefore, easily identifiable geochemically on the background of host MgO-poor and TiO$_2$-rich quartz gabbro (Fig. 5).

Again, the only other rocks in this traverse that have similar composition in terms of MgO and contains blebs of sulphides are Sublayer norites that occur at the very base of the SIC (Fig. 5). Finally, and, perhaps, most unexpectedly, one small melanorite body (up to 7 wt.% MgO) has been found to occur at the very roof of the SIC between underlying granophyre and overlying Onaping Formation in the Creighton area (Figs. 1 and 2d). Unlike melanorite bodies that are sitting in the interior of the SIC, it is distinguished by much finer-grained texture and high aspect ratios (length/width) of plagioclase (Fig. 6) implying a rapid cooling rate. Such discrete bodies of melanorites may be found in the future in the interior of the granophyre zone and in the North and East Range of the SIC (Fig. 1a).

**Melanorite bodies are an integral part of the SIC**. The chemical and mineralogical similarities of the melanorite bodies to Sublayer norites suggest that they are an integral part of the SIC. This is also strongly supported by the fact that the melanorite bodies are isotopically and geochemically almost indistinguishable from the host rocks of the SIC (Fig. 7). In particular, the geochemical affinity of the melanorite bodies to SIC norite, quartz gabbro and granophyre is evident from the incompatible element ratios (e.g., La/Ga), which are effectively constant in all of these rocks (Fig. 7a). They also have similar primitive mantle-normalized trace element concentration patterns, which show prominent negative Nb, Ta, P, Ti and Lu anomalies (Fig. 7c, d) and their initial $^{143}Nd/^{144}Nd$ compositions are identical, within error, to

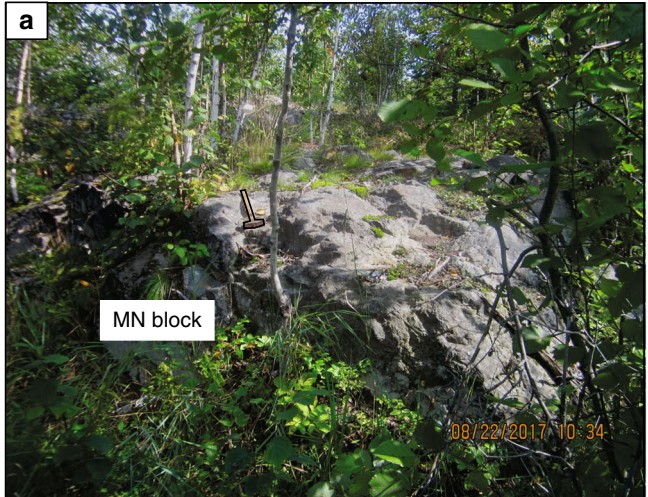

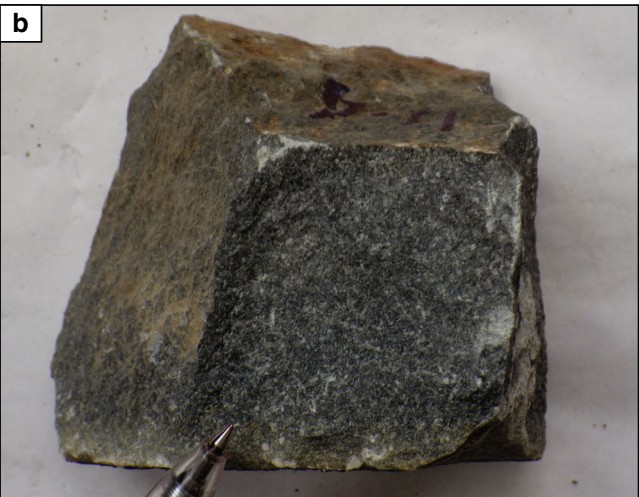

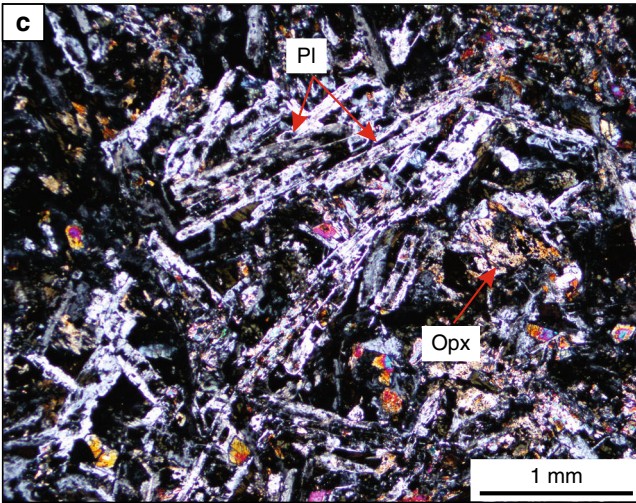

**Fig. 6** Melanorite body from the upper contact of the Creighton traverse in the South Range of the SIC. **a** Field photograph of an outcrop of Melanorite body that occurs along the contact of the SIC with Onaping Formation. The length of the hammer used for scale is 45 cm. Photo courtesy by Walter Peredery. **b** Photograph of a hand specimen of melanorite. The length of pencil used for scale is 2 cm. **c** Photomicrograph showing variable alteration of highly elongated plagioclase and small subhedral orthopyroxene. Sample W-17 in in cross-polarized light

the other lithologies of the SIC[37,38] (Fig. 7b). Some minor differences between the melanocratic bodies and host noritic rocks in the relative abundance of incompatible elements (e.g., REE) can probably be attributed to a variable amount of interstitial liquid trapped in these rocks.

### Discussion

A fundamental prediction of all the models that postulate the separation of the SIC into two melt layers[15–19,23] is that (mela) noritic rocks crystallized from the base upwards but not from roof downwards. Only granophyre is supposed to grow at the roof from the upper felsic melt layer. The occurrence of fine-grained melanorite body along the roof of the SIC (Figs. 1, 2d and 6) is thus not compatible with such models. The same is true for the melanorite bodies located in the interior of the SIC. These cannot be attributed to the incorporation into higher levels of the impact melt sheet of neither blocks of mafic target rocks from the basement during an impact[17] nor autobrecciated autoliths of solidified Sublayer norites. Such fragments of mafic rocks would have been denser (2.80–2.85 g/cm³)[2,18] than the granodioritic impact melt (2.47–2.50 g/cm³)[6] and would, therefore, sink, rather than float within it. In addition, the isotopic and geochemical similarities of the melanorite bodies to the hosts rocks (Fig. 7) indicate that these bodies are not xenolithic in origin but belong to intrusive rocks of the SIC. Attempts to explain these bodies in terms of emplacement of impact melt pulses from neighbouring sub-chambers of the SIC[39], mixing between adjacent, km-sized convective cells that show differences in degree and/or pathway of differentiation[36], or an upward trend towards more primitive MgO-rich composition of rocks[2] (Fig. 1b) are also problematic. These models imply that the melanorites should form (semi) continuous layers in the stratigraphy of the SIC, and are, therefore, at odds with the discrete, blocky shape of the melanorite occurrences (Fig. 2).

We conclude that if the large and dense blocks of melanorite could not physically be transferred from below, then the only remaining option is to derive them from above, i.e., by gravity settling from the roof of the SIC. It is inferred that the melanorite bodies are fragments (autoliths) of an original rock sequence of melanoritic rocks that crystallized from the top of the melt sheet downwards, but was disrupted and collapsed onto the temporary floor. Although the roof contact is poorly exposed, we have already identified one site where melanoritic rocks are located close to the roof of the SIC. This small outcrop (~5–7 m, Fig. 2d) is composed of fine-grained melanorite (20 vol. % orthopyroxene; 7 wt.% MgO), which has a striking similarity in terms of their major and trace element compositions (Fig. 7a, d) to the melanorite blocks and the mafic norites overlying the Sublayer, providing direct evidence for an original melanoritic roof sequence. The simultaneous crystallization of (mela)norite from the roof and base of the SIC may have only happened if the entire impact melt sheet was originally homogeneous[2–6]. If so, it suggests that the spectacular magmatic stratigraphy of the SIC (Fig. 1b) may only reflect its large-scale internal differentiation[2–6]. Our interpretation of melanorite blocks may seem aggrandized, as no one has ever surmised the prior existence of a roof noritic sequence in the SIC. We would like to stress, however—quite to the contrary —that it is the lack of a roof noritic sequence that should be regarded as an overlooked enigma of the SIC[40,41]. The SIC is essentially a shallow-level melt body whose predominant cooling from above was facilitated by cold ocean water circulating through the overlying breccia of Onaping Formation[40–42]. This cooling must have almost inevitably resulted in top-down crystallization of the impact melt sheet, coevally with bottom-up crystallization. Such inward crystallization is common for most

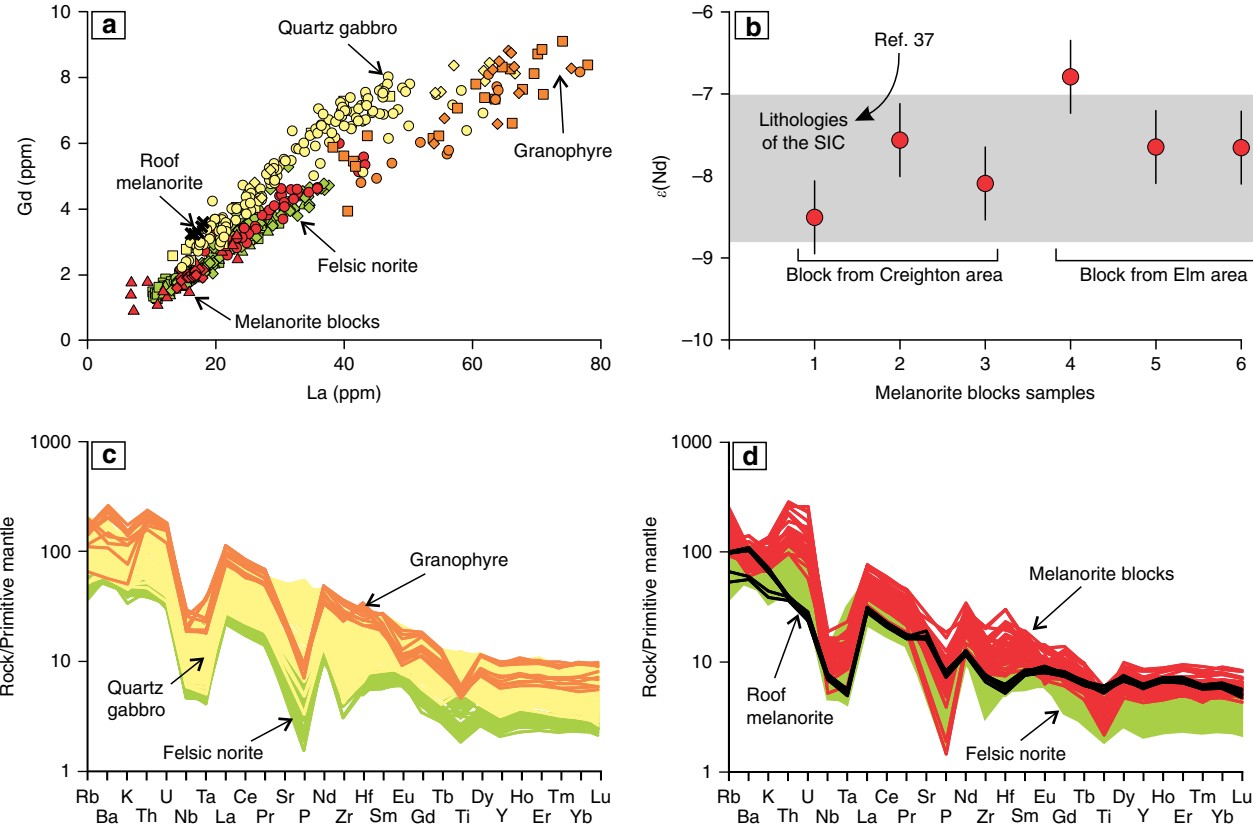

**Fig. 7** Plots illustrating chemical and isotopic similarity of the melanorite blocks to rocks composing the SIC. **a** Whole-rock variation of La versus Gd for the Elm, Creighton and Lockerby traverses, South Range showing that the ratio of these incompatible elements in the melanorite bodies, including the roof one, is indistinguishable from all other rocks of the SIC. Representative data are located in Supplementary Data 1. **b** The initial $\varepsilon_{Nd}$ values in two melanorite bodies from Elm and Creighton areas are identical within error to those reported from all other intrusive rocks of the SIC[37]. The bars represent the standard error of the mean. Original Nd isotopic data are located in Supplementary Data 2. **c** Primitive mantle-normalized diagram showing similar patterns of trace element concentrations of felsic norite, quartz gabbro and granophyre of the SIC from the Elm traverse, South Range. Original data are located in Supplementary Data 1. **d** Primitive mantle-normalized diagram showing similar patterns of trace element concentrations of the melanorite blocks and norites of the SIC from the Elm travers, South Range. Also plotted are roof melanorite from the outcrop at the Creighton traverse, South Range (Fig. 2d)

shallow-level mafic sills (e.g., Palisades[43] or Basistoppen[44]) and even some deeper layered intrusions (e.g., Skaergaard[45] or Kiglapait[46]), and should, therefore, be expected for the near-surface SIC as well.

We thus reaffirm that a large-scale impact, ~1.85 billion years ago, produced the few km thick Sudbury melt sheet, which initially comprised a single layer of superheated and homogeneous melt of granodioritic composition (Fig. 8a, b). The initial homogeneity is indicated by similar ratios and patterns of incompatible elements in the different SIC rocks, including granophyre (Fig. 7a, c). Also indicative of a single melt system are a systematic increase in abundance of REE and the substantial uniformity in REE patterns between the Sudbury lithologies[4], remarkably homogeneous incompatible trace element ratios in the entire section (e.g., Ce/Yb, Th/Nd)[2] and homogeneity in Os, Pb, Sr and Nd isotope compositions[37,38]. Some local isotopic heterogeneity reflects differences in the ages of the target rocks[47]. Upon cooling, the impact layer started crystallizing inwards from upper and lower margins producing orthopyroxene-rich cumulate that subsequently evolves into melanorite owing to postcumulus crystallization of a large amount of plagioclase-rich interstitial liquid. Rapid cooling and crystallization of the melt along the contact with the cold breccia of the Onaping Formation resulted in the fine-grained texture of roof melanoritic varieties (Figs. 2d and 6). Blebs of sulphides dispersed in melanorites (Figs. 3 and 4) indicate that the impact melt has likely reached

sulphide liquid immiscibility before onset of its crystallization. With subsequent arrival of liquidus plagioclase, the orthopyroxene + plagioclase cumulates started to crystallize into felsic norite (Fig. 8c).

We speculate that, if preserved intact, the overall compositional structure of the SIC would be grossly similar to that of the Skaergaard intrusion[45], i.e., it would consist of the Layered and Upper Border Series, growing contemporaneously from the roof and floor, exhibiting similar trends of phase crystallization and meeting at the Sandwich Horizon. The Upper Border Series would be about six to seven times thinner than the Layered Series because vigorous thermal convection in the main mass of melt will keep the upper thermal boundary layer thinner than the lower one[48]. However, such a structure is notably absent from the SIC (Fig. 1b). The most likely reason for this is that the roof sequence of melanorite/felsic norite was not stable and tectonic activity associated with late-stage crater adjustments resulted in their partial dislodgment and collapse, as discrete blocks, on the upwards growing chamber floor (Fig. 8c). The location of melanorite blocks at various stratigraphic levels in the SIC (Figs. 1 and 2) indicates several episodes of roof destruction. The process was still on-going during the formation of quartz gabbro, as indicated from the occurrence of, at least, one melanorite block in these rocks (Figs. 1b and 2c). It is not inconceivable that further searches will result in finding these blocks in the interior of the granophyre as well. This would indicate that the destruction

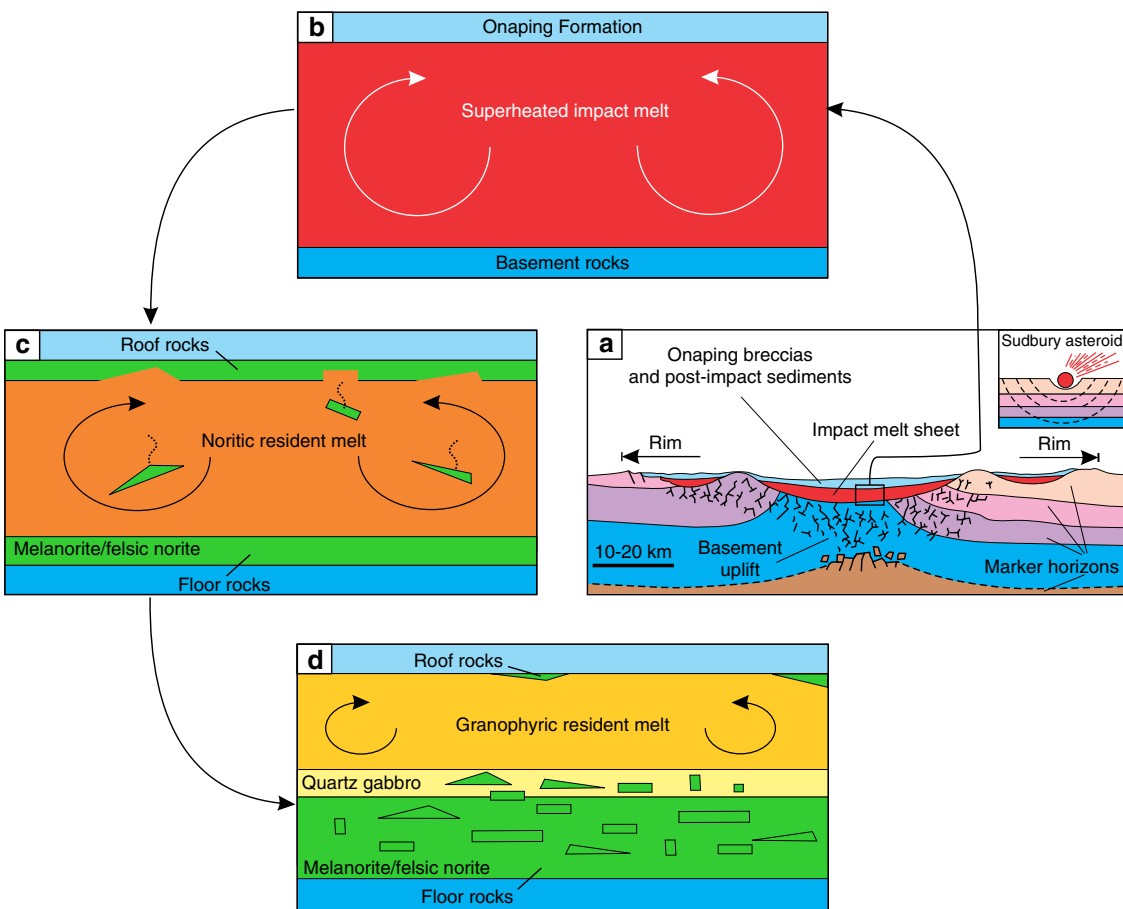

**Fig. 8** Model for the proposed crystallization history of the SIC. **a** A large asteroid hit the Earth (insert) and in a matter of a few minutes generated one layer of homogeneous and superheated impact melt covered by an ejecta layer. Modified from ref. [2]. **b** A single, homogeneous layer of superheated impact melt is sandwiched between the shocked basement rocks and overlying fall-back breccia of Onaping Formation. **c** After cooling, the impact melt starts crystallizing melanorite/felsic norite from all margins inwards. Tectonic activity results in the occasional disruption of the gravitationally-unstable roof sequence of melanorite/felsic norite and it collapses as angular blocks on the temporary chamber floor. **d** The process of roof sequence disruption continues during crystallization of quartz gabbro and by the time of granophyre crystallization, almost the entire roof sequence was destroyed and contributed substantially to the total sequence of the floor cumulates that must be full of melanorite/norite autoliths of various sizes. Only a few of these autoliths, referred in this study to as melanorite bodies, are currently documented (Figs. 1 and 2). The remnants of the original roof sequence of melanoritic composition are still locally preserved along the top contact of the SIC (Fig. 2d)

lasted throughout almost the entire crystallization interval of the impact melt sheet. This resulted in nearly complete disappearance of the roof sequence so that the residual felsic liquid came into direct contact with the overlying Onaping Formation (Fig. 8d). The overall thickness of the original noritic roof sequence is difficult to estimate due to its poor preservation. The size of the largest block (Fig. 2c) indicates, however, that the sequence was at least, 400–500 m thick. A detailed re-mapping of the upper part of the SIC is required to evaluate how much of the original noritic sequence is still preserved along the roof. What is clear, however, is that autoliths of the roof sequence may be quite abundant in the SIC and may substantially contribute to the overall thickness of noritic/gabbroic zones by forming ~15 vol. % of the sequence (i.e., 500 m out of 3000–3500 m). One of the future challenges is to define a clear set of textural and chemical criteria, expressly to map these autoliths in the field and laboratory.

Our study thus provides the long-missing field-based evidence to resolve the dilemma regarding the origin of the conspicuous cumulate stratigraphy of the SIC. The data strongly support the Sudbury impact resulting in complete mixing of the melted target rocks to form a homogeneous impact melt sheet and that this melt sheet then underwent effective differentiation to form a prominent compositionally layered body[2–6]. The conclusion is

fully consistent with results of previous phase-equilibria modelling indicating that crystallization sequence, mineral compositional trends and geochemical variations of the SIC can be adequately reproduced by fractional crystallization of a single batch of granodioritic magma[5]. If even such relatively small-sized melt sheets of quite evolved composition, as the SIC, can undergo differentiation and produce compositionally stratified bodies, then it is logical to assume that this will also occur in much larger terrestrial impact melt sheets of more primitive, mafic/ultramafic composition. The conclusion has potentially important implication with respect to the Hadean era (>3.8 billion years) of Earth evolution. This is the era, when the large multi-ring basins formed on the Moon. An even more massive bombardment must have also occurred on the Earth. With adjustments due to the larger gravitational cross-section, higher impact velocity and planetary gravity on Earth, estimates are approximately 3000 impact craters with diameters >100 km, with the largest impact basin being on the sub-continental scale[24,25,49]. Models of this bombardment and its effects[25,49,50] vary in sophistication but all result in a Hadean Earth where 100% of the surface was covered by impact ejecta and some ~10% by pools of impact melt, with a cumulative volume of ~$10^{11-12}$ km$^3$. Felsic lithologies in the Hadean proto-crust of the Earth are generally considered to form

through deep recycling of primary basaltic rocks by various geodynamic processes[27–32]. This study, however, raises the possibility that the considerable volumes of felsic rocks can be generated at a near-surface environment in the Hadean, through the large-scale differentiation of these voluminous impact melt sheets. Modelling of such a process shows the potential to produce up to $10^{8-9}$ km$^3$ of compositionally evolved impact melt (comparable to the volume of the present-day continental crust) in large Hadean basins[25].

While the differentiated Hadean impact melt pools are expected to have the same inherent density as the precursor target rocks of the Hadean proto-crust, the potential delamination[31,51,52] of their lower ultramafic portions could have produced residual blocks of more buoyant, mafic-to-felsic rocks. These blocks could potentially have played a role in the onset and development of the Hadean proto-continental crust[24–26]. We have employed here an original approach by Vaughan et al.[10] to illustrate this idea using the crystallization sequences in a pseudoternary phase diagram Fo-An-Qtz (Fig. 9) supplemented by stage-by-stage development of three successive generations of impact melt sheets (Fig. 10). The Hadean proto-crust is considered to be of mafic/ultramafic composition[26,53]. Its exact composition is unknown and, therefore, for illustration purposes, we use here a hypothetical composition X consisting of 60 wt.% enstatite, 20 wt.% forsterite and 20 wt.% anorthite (Figs. 9a and 10a). Its early melting by bolide impacts followed by igneous differentiation of impact melt sheets along a path olivine, orthopyroxene, orthopyroxene + plagioclase and orthopyroxene + plagioclase + quartz (+clinopyroxene) would produce stratified bodies (Figs. 9b and 10b–c) similar to those of well-known differentiated plutons (e.g. Stillwater and Bushveld Complexes)[54]. In the earlier-formed layered bodies, the ultramafic cumulates (dunite and orthopyroxenite) will predominate over the mafic-to-felsic ones (norite, quartz gabbro, diorite and granite). The dense ultramafic cumulates ($\rho = 3.25$ g/cm$^3$ for dunite, and $\rho = 3.20$ g/cm$^3$ for orthopyroxenite)[10] will be gravitationally unstable within the lighter Hadean proto-crust ($\rho \sim 3.12$ g/cm$^3$). They may therefore dislodge from the complexes and founder in the hot and plastically deformed rocks of the Hadean proto-crust to spread along the paleo Moho[51]. Multiple re-melting of the Hadean proto-crust together with the mafic-to-felsic cumulates from the preceding melting/crystallization events—expressed graphically as a mixing line between these two end-members (Fig. 9c, e)—would give rise to impact melt sheets with increasingly more evolved composition. Their large-scale differentiation would, therefore, produce stratified bodies with progressively larger proportions of mafic-to-felsic cumulates relative to ultramafic ones (Figs. 9d, e; 10d–i). The foundering of dense ultramafic cumulates would result in accumulation of substantial volumes of mafic-to-felsic rocks (buoyant crustal blocks) in the Hadean proto-crust making it compositionally layered and increasingly more evolved from the base towards the Earth's surface (Fig. 10a, d, g).

The impact melt sheets are supposed to produce rocks with geochemical features that are characteristic of shallow-level fractional crystallization (e.g. little fractionation of light from heavy REEs and pronounced negative Eu anomalies) of mafic/ultramafic melts in which the low-pressure minerals, such as plagioclase, were important constituents of the fractionating assemblage. This process cannot therefore be responsible for the tonalite–trondhjemite–granodiorite (TTG) suite that dominates the later formed Archaean (<3.8 billion years) crust because these have trace element chemistry (garnet signature) indicating their origin by high pressure geodynamic processes[31,32]. This prediction is difficult to rigorously test because the Hadean proto-crust was almost completely destroyed during the Moon-forming event

when the Earth was impacted by a Mars-sized object[55]. However, what has survived on the Earth from the Hadean proto-crust appears to be in line with our scenario. In particular, the magnetite-rich gneisses of the Acasta Gneiss Complex, northwest Canada—the only known felsic rocks of Hadean age (~4.02 billion years)—show chemical compositions distinctly different from rocks of the TTG suite. Their chemistry indicates that garnet was not involved in magma genesis or evolution and, instead, plagioclase was a major fractionating mineral[56]. These felsic gneisses were therefore attributed to such low-pressure processes as shallow-level fractionation of basaltic magma combined with assimilation of hydrothermally altered oceanic crust[56] and near-surface partial melting of hydrated basaltic rocks by asteroid impacts[26]. Our petrogenetic model is yet another possibility. In some respect, it shares features of both approaches—it implies the impact-induced bulk melting of target mafic/ultramafic rocks followed by shallow-level fractionation of impact melt sheets. Another piece of evidence supporting our model comes from the overlapping of Ti-in-zircon thermometry of Hadean detrital zircons (~4.4. billion years) from Archean metasedimentary rocks with those from granophyre of the SIC. This similarity testifies for the possible generation of more felsic lithologies within differentiated Hadean impact melt sheets caused by the intense asteroid bombardment of an early, hydrosphere-covered proto-crust[57]. We thus concur with an idea that most felsic rocks during the late Hadean were likely generated through impact-induced melting of the proto-crust[26] followed by shallow-level differentiation in impact melt sheets.

Finally, our study contributes to resolution of a current debate regarding the efficiency of internal differentiation in the Moon's impact melt sheets[10–12,14]. It supports a concept that such giant melt sheets, as 50 km thick and the 2500 km in diameter South Pole-Aitken[11], were able to undergo large-scale igneous differentiation and may, therefore, have contributed to the diversification of melt lithologies in the Moon's early crust. In particular, the formation of shallow noritic layers enriched in FeO and TiO$_2$ in the South Pole-Aitken basin could be attributed to its large-scale differentiation of its voluminous impact melt sheet[11,12,58]. It may well be, however, that only the thickest impact melt sheets on the Moon, i.e., within the multi-ring basins, had the potential to differentiate. This is because melt differentiation is favoured, among other factors (melt composition, sheet thickness, etc.), by high planetary gravity. The lower gravity environment of the Moon compared to the Earth[59] would, therefore, require thicker melt sheets (for a given composition) before they can differentiate, compared to their terrestrial counterparts[60]. Measurement-based evidence (i.e., rock samples or remote sensing) for large-scale differentiation of lunar impact melt rocks is still quite limited. Direct sampling of impact lithologies showed that some 30–50% of the rocks from highland landing sites sampled during the Apollo missions are impact lithologies: breccias, melt rocks and glasses. The identified melt rocks are, however, generally aphanitic to fine grained and charged with lithic and mineral clasts. Although believed, in some cases, to be linked to the large-scale impacts that formed multi-ring basins on the Moon[61–63], they likely represent ejected melt material and not direct samples of the voluminous, interior impact melt sheets, such as the SIC. There is, however, one sample in the Apollo collection, a noritic anorthosite in breccia 67955, that is interpreted on geochemical grounds as a differentiated impact melt rock[64]. It is believed to represent a sample of a coherent and differentiated impact melt sheet from a 4.2 billion years basin-sized impact, which was sampled and transported as ejecta to the Apollo 16 site by the Imbrium impact event. Additional evidence is expected to be obtained from multi-ring basins of the Moon, which are currently considered as a high-priority future target for

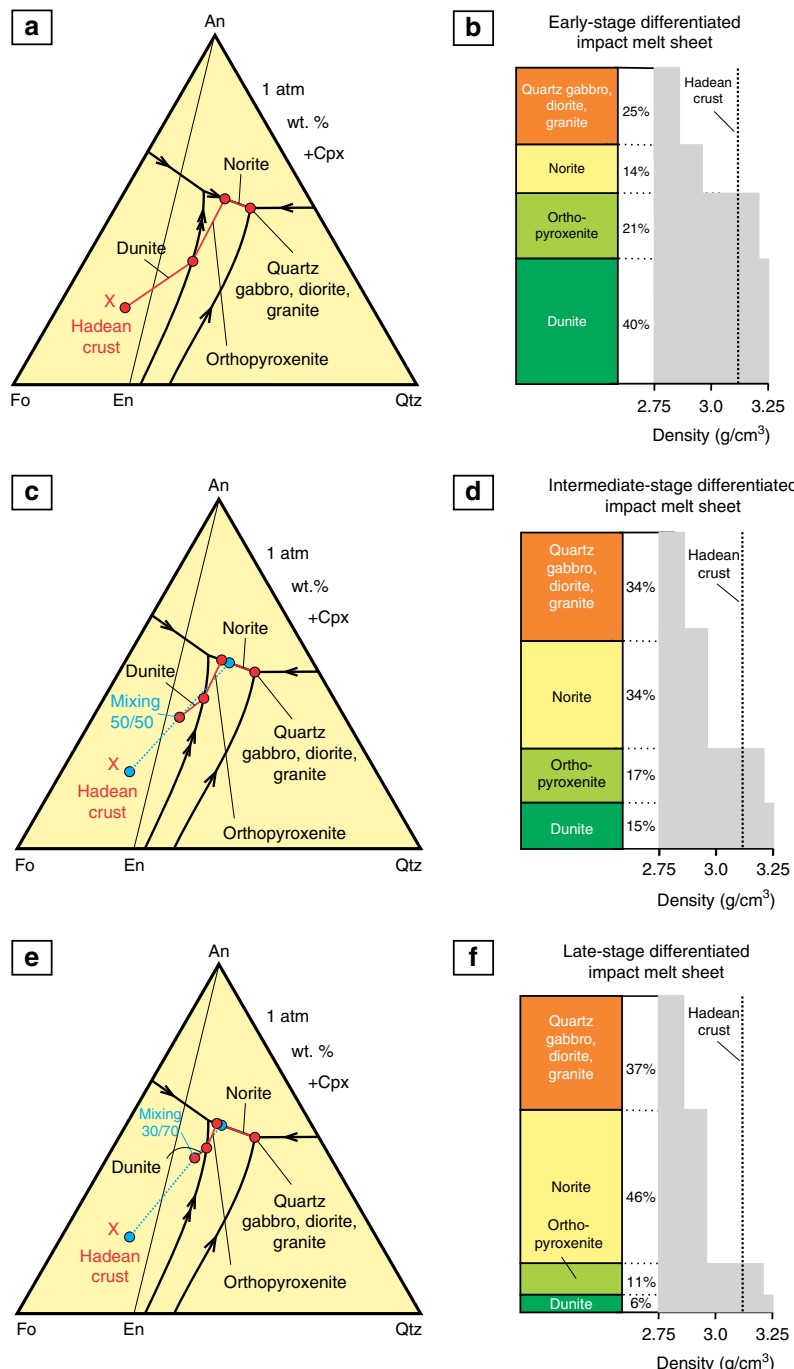

**Fig. 9** Fractional crystallization sequences in a pseudoternary phase diagram Fo-An-Qtz and model cumulate stratigraphies with related density profiles. **a**, **b** Low-pressure fractional crystallization of the early-stage impact melt sheets having a composition of the Hadean mafic crust (point X). In the resulting stratigraphy, the ultramafic cumulates (dunite and orthopyroxenite) substantially predominate over the mafic-to-felsic ones (norite, quartz gabbro, diorite and granite). **c**, **d** Low pressure fractional crystallization of the intermediate-stage impact melt sheets produced by melting of the 50/50 mixture of the Hadean mafic crust with the bulk of mafic-to-felsic cumulates from the early-stage impact melt sheets. In the resulting stratigraphy, mafic-to-felsic cumulates start prevailing over the ultramafic ones. **e**, **f** Low pressure fractional crystallization of the late-stage impact melt sheets produced by melting of the 30/70 mixture of the Hadean mafic crust with the bulk of mafic-to-felsic cumulates from the intermediate-stage impact melt sheets. In the resulting stratigraphy, mafic-to-felsic cumulates substantially predominate over the ultramafic ones. Crystallization sequences at **a**, **c** and **e** are indicated by red lines and solid dots. Mixing end-members—the Hadean proto-crust and the bulk of mafic-to-felsic cumulates of impact melt sheets—are indicated by blue dots connected by dotted blue lines. Mixing proportions are arbitrarily chosen. The weight proportions of cumulates shown next to the model stratigraphies are constrained using a lever rule directly from a phase diagram. Densities of cumulate rocks are from ref. [10]. Density of the Hadean proto-crust X is calculated from its hypothetical initial composition (60 wt.% En, 20 wt.% Fo and 20 wt.% An) using data from ref. [10]. Phase diagram is simplified from ref. [70]

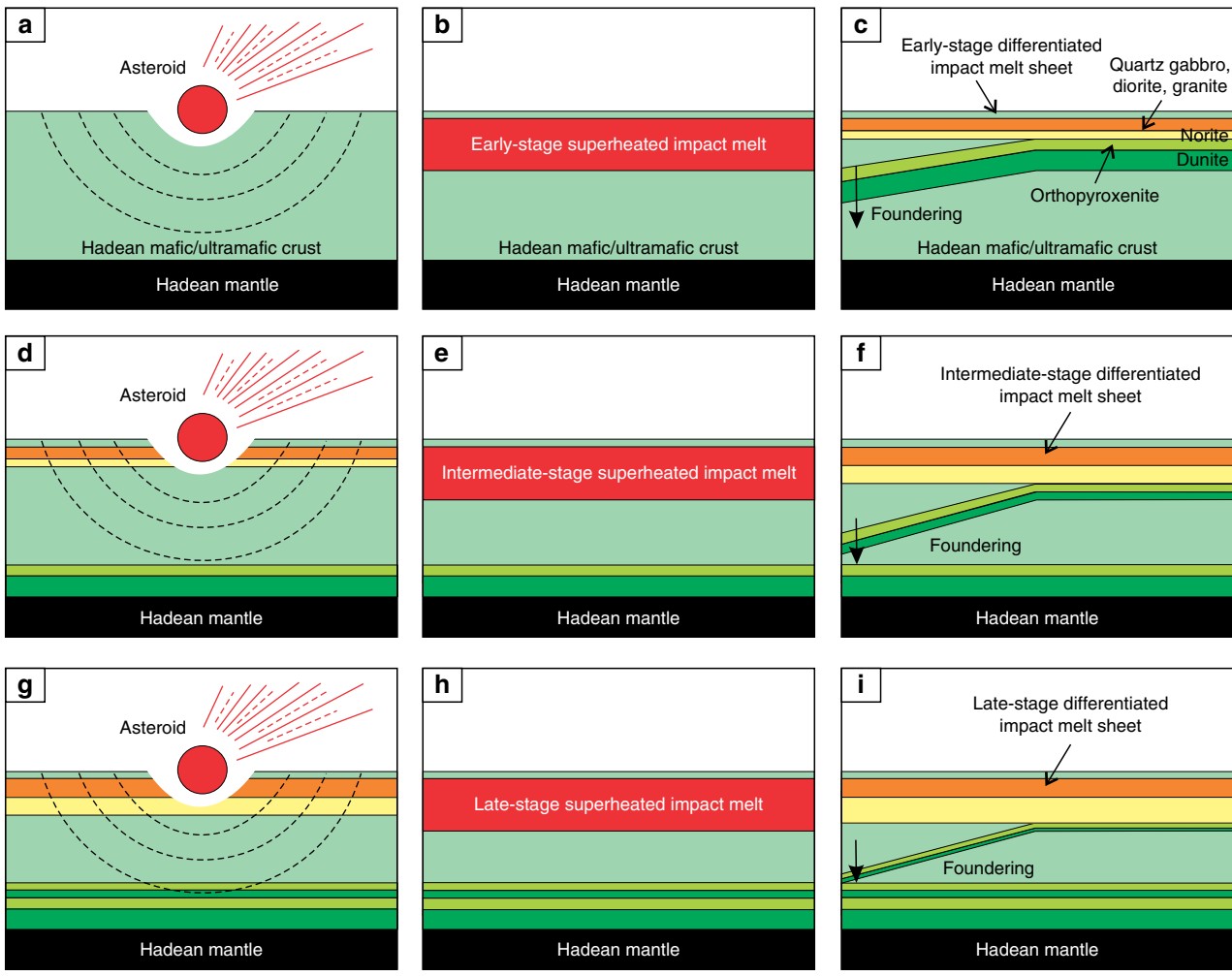

**Fig. 10** Model for the proposed compositional evolution of the Hadean mafic/ultramafic proto-crust due to fractional crystallization of impact melt sheets followed by delamination of their ultramafic portions. **a–c** Early Stage. A large asteroid caused melting of the Hadean proto-crust X (Fig. 9a) and generated a sheet of homogeneous and superheated impact melt. After cooling, the impact melt has differentiated into a well-stratified igneous body with ultramafic cumulates (dunite and orthopyroxenite) substantially predominating over the mafic-to-felsic ones (norite, quartz gabbro, diorite and granite). Ultramafic cumulates are denser than proto-crust (Fig. 9) and therefore they foundered towards the crust-mantle boundary[51,52]. Fractional crystallization sequence is shown in Fig. 9a, b; **d–f** Intermediate Stage. A large asteroid caused melting of the Hadean proto-crust together with mafic-to-felsic cumulates from the previous impact event and generated a layer of homogeneous and superheated impact melt. After cooling, the impact melt has differentiated into a well-stratified igneous body in which mafic-to-felsic cumulates started prevailing over the ultramafic ones. This is followed by foundering of dense ultramafic cumulates that resulted in enriching the proto-crust in compositionally evolved cumulates. Fractional crystallization sequence is shown in Fig. 9c, d; **g–i** Late Stage. A large asteroid caused melting of the Hadean proto-crust together with mafic-to-felsic cumulates from the previous impact event and generated a layer of homogeneous and superheated impact melt. After cooling, the impact melt has differentiated into a well-stratified igneous body in which mafic-to-felsic cumulates substantially predominate over the ultramafic ones. This is again followed by foundering of dense ultramafic cumulates that further enriches the proto-crust in compositionally evolved cumulates. Fractional crystallization sequence is shown in Fig. 9e, f. The ultimate result of this process is accumulation of the substantial volumes of mafic-to-felsic cumulates with low-pressure geochemical characteristics in the Hadean proto-crust[24,25] making it compositionally layered and increasingly more evolved from the base towards the Earth's surface. The cartoon is for illustration purposes only since neither the exact composition nor thickness of the Hadean proto-crust/impact melt sheets are currently known

human or robotic landed missions during which more impact rock samples will be collected for further analysis[65].

## Methods
**Rock sampling, major and trace chemical analyses**. Three detailed sections through the melanorite bodies and adjacent rocks of the SIC were sampled. Over 600 samples (each sampled weighted ~1 kg) were collected for petrographic, whole-rock (XRF and ICP MS) and mineral compositional analyses. Major and trace elements were analysed by Genalysis Intertek Laboratory Services in Australia. Major elements were determined through X-ray fluorescence analyses (XRF) of compressed powder pellets. Calibrations used the international rock standard SARM8 as well as in-house controls. Agreement with recommended values was better than 0.6% for $Cr_2O_3$, $Fe_2O_3$, MgO, $Al_2O_3$ and better than for 1–6% for all other major elements. Trace elements were determined through inductively-coupled plasma mass spectrometry and atomic emission spectrometry (ICP-MS/ICP-AES) with four acid digests. Each ICP-MS analysis was accompanied by control standards GTS-2a, AMIS0167, and AMIS0013 and selected samples were re-analyzed to check anomalous results. For all elements, the relative standard deviations were less than 10%. A complete list of analyses from the Elm traverse of the SIC (Figs. 5 and 7) is available online within the electronic Supplementary Data 1.

**Isotopic analyses**. The methods for Sm-Nd isotope analyses followed standard procedures at the Geological Survey of Finland[66]. The 120–400 mg of powdered sample was spiked with a $^{149}$Sm-$^{150}$Nd tracer. The sample-spike mixture was dissolved in HF-HNO$_3$ in Savillex screw-cap beakers on hot plate (mafic rocks) or in sealed Teflon bombs in an oven at 180 °C (felsic rocks) for 48 h. After careful evaporation of the fluorides, the residue was dissolved in 6 N HCl and a clear

solution was achieved. Sm and Nd were separated in two stages using a conventional cation exchange procedure (7 ml of AG50Wx8 ion exchange resin in a bed of 12 cm length) and a modified version of the Teflon-HDEHP (hydrogen diethylhexyl phosphate) method[67]. The measurements were made in a dynamic mode on a VG SECTOR 54 mass spectrometer using Ta-Re triple filaments. $^{143}Nd/^{144}Nd$ ratio is normalized to $^{146}Nd/^{144}Nd = 0.7219$. The average value for the La Jolla standard was $^{143}Nd/^{144}Nd = 0.511854 \pm 0.000008$ (1 standard deviation, $n = 27$). The Sm/Nd ratio of the spike has been calibrated against the Caltech mixed Sm/Nd standard[68]. Based on duplicated analyses, the error in $^{147}Sm/^{144}Nd$ is estimated to be 0.4%. Initial $^{143}Nd/^{144}Nd$ ratios and $\varepsilon_{Nd}$ values were calculated with the following parameters: $\lambda^{147}Sm = 6.54 \times 10^{-12}a^{-1}$, $^{147}Sm/^{144}Nd = 0.1966$ and $^{143}Nd/^{144}Nd = 0.51264$ for present CHUR. Depleted mantle model ages ($T_{DM}$) were calculated according to DePaolo's method[69]. Measurement on the rock standard BCR-1 provided the following values: Sm = 6.58 ppm, Nd = 28.8 ppm, 147Sm/144Nd = 0.1380, 143Nd/144Nd = $0.51264 \pm 0.00002$. Total procedural blank was <0.5 ng for Nd. A complete list of Sm-Nd analyses from the melanorite bodies of the SIC (Fig. 7) is available online within the electronic Supplementary Data 2.

## Data availability

The authors declare that all relevant data are available within the article and its supplementary information files.

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

## Acknowledgements

We are grateful to P. Lightfoot, M. Lesher, J. Mungall, W. Peredery, R. Keays, E. Pattison, P. Golightly, M. Andreoli, T. Ubide, E. Hunt, R. James, J. Day, U. Reimold, G. Osinski, J. Bédard, J. Kramers, N. Arndt, C. Hawkesworth and R. Gibson for fruitful discussions over the years on several aspects of this study, as well as for critical comments and useful suggestions on the earlier versions of this manuscript. The research was supported by several research grants to R.L and S.C from the National Research Foundation of South Africa and DST-NRF CIMERA.

## Author contributions

R.L. and S.C. undertook field work, mapping and geochemical sampling of the SIC as well as conceptualized the original idea and wrote a draft of the paper. R.G. has contributed with knowledge on bolide impacts and in extending the results of this research to the other terrestrial planets. H.H. performed whole-rock isotopic analyses and participated in data processing and interpretation. All co-authors discussed the results and problems and contributed to producing a final draft for peer reviews.

## Additional information

**Competing interests:** The authors declare no competing interests.

