## [Peer Review File · Nature Communications]

Reviewer #1 (Remarks to the Author):

This is a well written manuscript that addresses a long-standing conundrum concerning the origin of systematic compositional zoning with stratigraphic height in the Sudbury Igneous Complex. Thanks to the efforts of these authors, among others, it is now widely accepted that the SIC magma formed by impact melting about 1.8 billion years ago, but the detailed petrogenesis of the compositional variations have remained rather enigmatic, as nicely documented in the manuscript.

Here the authors present new observations that support an origin by fractional crystallization rather than other types of processes that have been proposed. The key new observation is the physical distribution of blocks of norite that appear to have crystallized on the roof of the SIC and then became dislodged and settled through the magma. Personally I have always been willing to accept a fractional crystallization model for the compositional zoning of the SIC, and these new observations strengthen that interpretation. So the conclusions about the petrologic and magmatic evolution of the SIC seem to be robust, and that part of the paper is well reasoned and clearly presented.

The problem that I have with this paper is the attempt to extrapolate to other planets and wrap the study in a broader significance about early planetary evolution. This aspect of the work is speculative, superficial, and not at all convincing. For example, although the authors provide some citations in line 8, they do not come to grips with the overwhelming lack of measurement-based (i.e., either samples or remote sensing) evidence for large-scale differentiation of lunar impact melt rocks (but see Norman et al 2016 GCA for an alternate example, not cited), or the fundamental physical and thermo-mechanical differences between the crust of the Earth vs. other terrestrial planets. This is especially important here considering the age and unique (for a large terrestrial impact structure) petro-tectonic setting of the SIC. Mars might provide a more apt analogy than the Moon but there is no evidence that I am aware of for strong impact-driven differentiation of Mars beyond possibly the earliest stages of planetary accretion (i.e., formation of the crustal dichotomy by a giant impact).

More specific to the Earth, it appears difficult to physically and chemically isolate only the evolved felsic differentiates of impact melt sheets like those produced in the SIC in a way that would influence the large-scale geochemical evolution of the depleted mantle and continental crust. So while it might be possible that Hadean impact melt sheets were more strongly differentiated than younger examples, the manuscript does not provide adequate support for the speculation that this process may have influenced the large-scale geodynamical evolution of the Earth as implied by lines 122-126.

My recommendation is that this manuscript is better suited to Geology, and that the paper would be strengthened if the discussion maintained its focus on the SIC, perhaps by addressing other unusual aspects such as the economic mineralization, rather than attempting to hype the study beyond what can be reasonably supported in a short-format article.

Some line-by-line comments are provided below.

Line 10: I am not completely convinced that the SIC is the best-preserved impact melt sheet on Earth. It might be the best exposed example, but it has been tectonically and hydrothermally altered. The Chicxulub or even Manicouagan melt sheets might actually be better preserved even if exposure is not as good as the SIC.

Line 19 and elsewhere: I am not sure that 'compositional stratigraphy' is the best term to describe the variations seen in the SIC. The term 'stratigraphy' typically implies discrete layers but as shown in Fig. 2, the compositional variation is smooth and systematic such that 'compositional zoning with stratigraphic height' might be a better description.

Line 21 and elsewhere: Personally, I would not describe the compositional variation of the SIC from about 1-7 wt% MgO as 'large-scale differentiation'. Certainly there are some more mafic compositions, but I don't see the evidence for compositionally extreme ultramafic or anorthositic units such as those in layered mafic intrusions such as Bushveld and Stillwater. The SIC seems to be moderately zoned in bulk compositions similar to a lava lake such as Kilauea Iki, rather than strongly differentiated

Line 28: replace 'size' with 'diameter'

Line 30: With respect to the authors, I just don't think the SIC plays as much role in people's thinking about impact melt sheets as it used to. We now see similar evidence for moderate compositional zoning in the melt sheet at Manicouagan, and the possibility that melt sheets can differentiate is now widely considered as a possibility (as shown by refs cited in the Introduction) even if the scale of that differentiation remains poorly established, especially on other planets with very different crustal compositions and thermal structures. So while the SIC remains a classic locality, I just don't think that the justification for using the SIC as a fundamental planetary analogue is quite as compelling as implied by the authors here.

Line 47: 'dramatic' is a bit of an over-statement here.

Line 119: As noted above, I would say that large melt sheets 'may' undergo crystal-melt differentiation rather than asserting this as a certainty. Although thermo-mechanical models for efficient differentiation of large melt sheets have been published, I am not aware of any that are differentiated to the extent of layered mafic intrusions such as Bushveld and Stillwater, which is what the authors are implying here. Even the study of an apparently differentiated lunar melt rock by Norman et al 2016 GCA proposed a much more modest process more analogous to filter pressing rather than crystal layering. The trace element variations in the SIC that are documented in Fig. 3 also seems more compatible with a more subdued style of crystal-melt segregation.

The figures are well drawn and nicely illustrate key observations of the study and relevant aspects of the SIC.

Reviewer #2 (Remarks to the Author):

Overview:

The discovery of enclaves of more melanocratic, more highly-magnesian norite in the upper layers of stratigraphy of the Sudbury Igneous Complex (SIC) represents a useful and important finding in one of the world's most distinctive and important igneous bodies. The explanation for these bodies does indeed pose important questions in terms of the evolution of impact melt sheets, as indicated by the authors. However, I believe that the authors have not done enough here to support their interpretation of in situ internal differentiation having produced melanorites at the roof. Specifically, the analysis of existing models is virtually absent, and the context of how these rocks relate to existing basal melanorite units, and how they would fit into a differentiation model for the rest of the SIC rocks is not really addressed. The nature of the physical evidence is not entirely clear (size and shape of enclaves is important, but precisely how this information was determined is not spelled out), and the manuscript is under-referenced and slightly casual with dimensions and age details. I would recommend revision with emphasis on attention to details such as these to maximize the potential impact of this work.

Detailed comments:

Line Comment

7 The citation superscript (1-8) should be attached to the word 'controversial', rather than the following word, 'whether'.

27, 28 Should read "10s of km", and "100s of km", or better still, "tens of kilometres" and "hundreds of kilometres".

31 Insert comma after "Earth"

33 The age of the SIC should be reported as ca. 1.85 Ga, rather than 1.8; I think 500 million years is worthy of quibbling about. The age also requires a citation, for which the 1984 Krogh et al.

source and/or Corfu & Lightfoot (1996) are still suitable; more recent efforts haven't changed anything.

34 The source of the melt sheet temperature data should also be cited; between Grieve, Prevec & Cawthorn, Ivanov, Ariskin, you have a few choices.

50 I think you should avoid "Opx" in the text, and replace with the word orthopyroxene. (In spite of Fig. 2 caption)

51 Similarly, reference to "Creighton, South Range", is overly brief; "in the Creighton area of the SIC South Range" would be more useful.

58 Rather than "no other melanorite bodies occur, as far as 1.5 km etc.", which reads awkwardly, I'd suggest: "No other melanorite bodies have been found within 1.5 km..."

69 Reference to possible sources of inherited/xenolithic mafic target rocks:

- First, the presence of any inherited target rocks lying around in the melt sheet after thermal equilibration/homogenization would require that the melt sheet was not capable of consuming those (large) fragments, and would contradict the assumption of a superheated melt sheet after those first few seconds/minutes, as it would, by definition, have frozen after trying and failing to consume those xenoliths. It is generally accepted that there were no large target rock fragments left after melt equilibration.

- Second, there are other much better sources of ultramafic rock in the footwall, specifically ultramafic bodies hosted in the North Range footwall both near (Fraser; see Lightfoot) and distal to (Smith et al. 2014) the SIC contact. The East Bull Lake suite are mainly leucogabbronorites (see published work by James, Peck & friends), with notoriously little ultramafic component.

- I concur that for these reasons, as well as those you mention (geochemical consistency of your enclaves with other SIC rocks), the enclaves are unlikely to be xenolithic, but you should/can present a much more robust argument for this.

77 The comment that the previously recognized melanorite is 3 km below your reported findings is rather loose with the data; the SIC melt sheet is approximately 2.5 km thick now, preserving lithologies from base to roof, so while there have been suggestions that the melt may have been more extensive/thicker (up to 4.5 km by Ariskin et al., 1999, and Lightfoot (2017) gives the thickness somewhat unhelpfully as 1.5-5 km, depending on variations in floor topography, and his Fig 3.34 addressing this specific topic is not dimensionally quantified, vertically), this is not really relevant to stratigraphic references relating to the observed sequence. A number such as 2 km would still make your point perfectly well, and be consistent with observed stratigraphic constraints.

77-78 A statement such as “Clearly it is physically impossible to relocate such large bodies stratigraphically up-section” is not a useful statement, with no accompanying analysis or citation to sources who have done so. As it is, this comprises your rejection of at least some of the proposed models, with no substantiated critical analysis to accompany it.

For example, the authors have cited Golightly’s suggestion of segregating mafic and felsic components in the post-impact melt phase, a model which superficially would explain bits of mafic material trapped in felsic host, as well as, presumably, bits of felsic material trapped in mafic rocks (which are arguably present in the South Range as well). I believe vigorous convection within the cooling superheated melt sheet combined with autobrecciation of earlier cumulate material (such as the basal melanorites) is also an option. Neither of these are discussed or ruled out here, which weakens the proposed interpretation. Some of this refutation depends on the available physical data (size & shape of enclaves), which is commented on further below.

88 The reference to the Palisades Sill should be accompanied by a citation to a source (Naslund, for example?).

93 See line 50 comment

97 See line 33 comment

111 500 m of roof melanorite would not represent 10% of the SIC, which has never been proposed as a 5 km thick entity throughout(double its current known thickness); how about “at least 5%”, which leaves it ‘significant but open to growth’?

285 Insert comma after South Range.

Systemic concerns:

More rigorous analysis of genetic models needed:

As mentioned above (line 77-78), the findings of melanorite in the upper/uppermost SIC melt sheet stratigraphy are potentially explicable by proposed models which have not been refuted in any critical fashion. Specifically:

- Golightly’s model would presumably require that the felsic & mafic components were “clast-melt mixtures” (which is probably not consistent with thermal equilibration models in which all clasts should be consumed if prolonged superheat is to be accepted; if it’s not, then this is not a

problem) which floated to the top and bottom according to gross density of the mixture. In this case, however, the clasts might be expected to be unconsumed target rock lithologies (this is not explicitly stated by Golightly, but seems to be implicit), which you have noted is unlikely (see also comment for line 69).

- Convective brecciation of early roof & floor rocks would be conceivable, with the roof & floor (rather than the hotter interior) being the most likely places to later preserve these trapped blocks. The environment of a superheated melt sheet contained within a steep thermal gradient is as good a place to facilitate vigorous convection as you could ask for in nature, so some density modelling/calculations to illustrate the potential problems with moving very large fragments within the melt would be required in order to convincingly dismiss such a model.

A large part (probably more important to the argument than the geochemistry) of the author's argument for the proximal/overlying origins for these enclaves is their angular shape (which mitigates against an immiscibility origin, for example) and their large size (which mitigates against significant transport distances upwards against gravity); however, you have explicitly stated (lines 136-138) that the fragments are extremely difficult to identify in the field or even petrographically, and that geochemical criteria are the only reliable fingerprints for them. Your argument nonetheless uses their size and shape as critical parameters, even though your sampling density does not allow for the delineation or confirmation of the proposed dimensions of the fragments as shown in Fig. 1. The examples provided in the Extended Data, such as Fig. 2b, showing fist sized fragments of more melanocratic rock, are fine as examples of xenolithic fragments, but not itself convincing as relates to much larger blocks, for which the Fig 2 a photo is not a useful illustration in terms of elucidating lithological contrast. This applies similarly to Fig 1 and 3, where the ability to define the dimensions of the enclaves is not clear. However, the photos do appear to show that you are in fact able to distinguish the darker coloured xenoliths from their surroundings, which provides credibility to your size and shape estimates; this point needs to be made clear, as it appears to contradict the statements cited earlier in this paragraph.

The sample shown in Fig 3 appears to be very rapidly cooled in contrast to those in fig 1 and 2, which look like 'normal' SIC rocks; the sample in Fig 3 is anomalous, and appears to be dyke-like or nearly chilled. This petrographic contrast is noted (line 149) but not discussed.

Consistency with context missing:

The authors may appear to be proposing, then, that the earliest material to have crystallised in the melt sheet was of melanocratic composition. If this is meant to be representative of early liquids, this would be entirely inconsistent with existing models which correlate the bulk composition of the impact target rocks with the compositions of early melt injections into the floor (i.e., Offset dykes) and more directly still with lowermost units of the fallback breccia (Onaping Fm), all of which are quartz dioritic in broad composition, and neither of which are melanocratic nor Mg-rich.

Alternatively, and more likely, the authors are proposing that the SIC cooled and fractionated inwards consistent with the closed system crystallization of a “normal” intrusion or sill, such as the Skaergaard Intrusion (Greenland). This would require an inwards trending, vaguely symmetrical geochemical profile consistent with internal differentiation, which is not obviously the case here (as shown in their Fig 2, for example). I am not suggesting that this case cannot be made on the basis of existing data, but no effort has been made to present such a case here.

In addition, the term melanorite is used very loosely in the manuscript as it relates to existing SIC nomenclature. The basal unit of the melt sheet has been variously named as the Mafic Norite in the North Range, and the Quartz Norite in the South Range, each distinguished by petrographic and compositional peculiarities from each other, and from the overlying so-called Felsic Norite (cf. Lightfoot; various). If we lump the Mafic and Qz Norites together for ease of discussion, this unit is a discontinuous/intermittent basal unit overlying contact sublayer embayments, which comprise basal breccias. The implication is that the more melanocratic norite overlying the embayments is then locally-formed, or perhaps residual opx-cumulate norite which has been eroded away where not protected by the embayment mouth structures. It appears to be cut/intruded by the underlying sublayer, which would be consistent with any of the proposed models. It is not made at all clear how the authors are defining their term melanorite in this context.

Finally, the last paragraph of the manuscript (lines 113-128) are devoted to the potential significance of these findings as they relate to Hadean crustal evolution, specifically the concept that early impact melts could be generating felsic crust. This discussion does not qualify how the proposed model of internal differentiation would differ from any other model proposed for Sudbury which also involve the production of differentiated mafic and felsic melts/rocks from an impact process (i.e., the process may differ, but the outcomes are the same; how does this help us in the Hadean? This needs to be made clear.). Since the SIC melt was a product of homogenization of already very felsic (largely granitic, s.l.) target rocks, it's relevance to making felsic rocks out of impacts into primitive, possibly ultramafic Hadean oceanic crust, is also unclear.

Reviewer #3 (Remarks to the Author):

The paper's thesis, that impact melt sheets were an important geologic product on the early Earth and that differentiation of them was likely and, thus, important in the evolution of Earth's crust, is a finding that should shape any geologist's evaluation of early Earth processes. For that reason, the paper is a compelling contribution of broad interest. I append comments that are mostly minor and intended to help the authors polish the submitted paper. The most important comments are (9), (10), and (12). I suspect those comments can be addressed without much work and that the paper can be published soon thereafter.

1) Opening paragraph. The SIC is described as the “best preserved impact melt sheet on Earth.” The Chicxulub melt sheet may be better preserved, although it is buried. The SIC is dramatic because it is “exposed” and “accessible.” Minor rewording may be appropriate to better convey the status of SIC.

2) Opening paragraph. For this journal’s audience, it will be important to define “melanorite.”

3) Opening paragraph. The description of Hadean impact melt sheets and the implied effect on Earth’s crust should probably cite a paper by Grieve, 1980, in Precambrian Research.

4) Opening paragraph. Here or elsewhere, the paper may need to refer to other work to convince readers that impact melt sheets may be an abundant geologic product of the Hadean. A relatively recent example was published by Abramov and others, 2013, in *Chemie der Erde*. That paper or the references therein may be suitable. There was another recent paper by Marchi and others that appeared in *Nature*, *Science*, or one of their recent offspring.

5) Second paragraph. The opening line is important. The geologic process of impact cratering was a major process affecting Hadean Earth. It is a process ignored by most investigators of the Hadean. The current manuscript may fundamentally affect the way geologists view the early Earth.

6) Line 31: The SIC is dramatic because it is “exposed” and “accessible.” Minor rewording may be appropriate to better convey the status of SIC.

7) Lines 34 through 36. Are the commonly-used lithological names (norite, granophyre, and quartz gabbro) strictly correct? If not, it may be better to provide readers with both the historical terminology and petrologically-correct terminology, so that readers are not compositionally misled.

8) Line 47: The word “dramatic” may be inappropriate.

9) Lines 49-51: The observed properties (medium-grained, sulphide-bearing melanocratic norite) is a clue that prompts a question. Is the sulphide consistent with the model proposed in this paper? Sudbury sulphide has two general sources: (a) it settles to the bottom of the melt sheet due to differentiation and (b) is produced hydrothermally during the solidification and cooling of the melt sheet. In the case of (a), one wonders how it could be trapped in a type of rock that formed at the top of the melt sheet (lines 81-82) and/or in rock that sometimes still occurs near the top of the melt sheet (Figure 1d). In the case of (b), why would the sulphide preferentially impregnate the melanorite blocks? An explanation of the sulphide is needed that is consistent with the paper’s model.

10)Line 67: The description of “angular blocks” would be compelling evidence of the model proposed in the paper. Yet, evidence of “angular” morphology seems weak. In the “Methods” section (lines 136-140), readers are told “visual identification of the melanorite bodies in the field is extremely difficult.....Geochemistry appears to be the only tool that allows the confident identification and mapping of melanorite bodies in the SIC.” If they are not confidently mapped in the field, how confident is the assessment of an “angular” morphology? Can the authors clarify?

11)Lines 69-70: The isotopic and geochemical evidence of an SIC origin of the blocks is compelling.

12)Lines 81-82: Did the SIC have sloping walls? If so, could the melanorite in the clasts have solidified high up on the basin walls (which is still effectively the base of the melt sheet and, thus, likely to contain sulphide) and then slumped into the melt sheet?

13)Line 85: What evidence exists for SIC cooling via circulating ocean water? While the papers cited may describe models for the Onaping that involve melt-seawater interaction, I do not recall the papers demonstrating (a) that seawater was the principal cooling mechanism of the underlying SIC or that (b) the sea was the dominant source of water flowing around and through SIC. The authors do not need to (a) and (b) to make their case. Thus, the simple solution is to simply delete the phrase “cold circulating ocean water.” It is sufficient to say that the top of the melt sheet (just like a lava lake or lava flow) cooled and solidified before the interior of the melt sheet.

14)Lines 117-120. The Vaughn and Head reference (PSS 2014) is fine. The Hurwitz and Kring reference (EPSL 2015) is not appropriate. I suspect the authors are confusing that paper with another (Hurwitz and Kring, JGR 2014).

15)Same lines. “South Pole Aitken” should be written “South Pole-Aitken.”

16)Same lines. This section is another good place for the Grieve (1980) reference described in note (3) above.

17)Same lines. The text overlooks the potential consequences of bulk melt composition. Does the ease with which differentiation occurs depend on melt composition? How does the Sudbury melt composition compare with those inferred for the Hadean or on other planetary bodies. Also, the text overlooks the potential consequences of planetary gravity. If I recall correctly, Paul Warren argued the propensity for differentiation depends on gravity, so melt sheet differentiation that occurs on the Earth may not necessarily occur in a melt sheet of the same size on another planetary body. This point is not central to the paper’s argument, but on how the paper’s result can be used elsewhere. It is, thus, a minor point that the authors can address with simple rewording.

18) Lines 126-128. The text seems to contradict a finding by Wielicki and others (EPSL 2012). The text refers to a paper by Kenny and others (Geology 2016). If the Geology paper demonstrated the older EPSL work was flawed, then the content of lines 126-128 is fine. If, however, a discrepancy between the Geology and EPSL papers remains, the authors should refer to both works.

19) Figure 4a. The morphology of the crater cross-section is probably incorrect. A central peak complex crater is illustrated. If Sudbury is as large as Chicxulub (or larger), then it had a peak-ring basin morphology like the well-exposed Schrodinger basin on the Moon. Grieve and others (JGR 1991) may have produced the first illustration of a reconstructed Sudbury, although there are other, more recent examples of peak-ring basin cross-sections.

Replies to reviewers' comments:

Reviewer #1 (Remarks to the Author):

A short summary of our changes/replies to the reviewer's comments:

The reviewer has raised three major issues:

1. Extrapolation of our study to the Earth's Hadean time. This part is now substantially expanded and includes the phase equilibria modelling of internal differentiation in compositionally primitive Hadean impact melt sheets. We also think that there is no difficulty in physical and chemical isolation of only the evolved felsic differentiates of these impact melt sheets to enrich the Hadean proto-crust in felsic material. This may happen in response to foundering of the dense ultramafic cumulates from these sheets as recently has been numerically modelled by Roman & Jaupart (2016) for layered intrusions of the Bushveld-type. (Roman & Jaupart. The fate of mafic and ultramafic intrusions in the continental crust. *Earth and Planetary Science Letters* 453, 131-140). We have now added two additional figures to the paper to illustrate these ideas.

2. Extrapolation of our study to other planets. We have now added a section on the Moon to our paper to address this issue. No data are available for the Mars at the moment to discuss them in the paper.

3. Limited degree of differentiation in the SIC. This issue is repeatedly raised below. We feel that a reviewer is missing here one important point. Yes, this is true that the SIC does not show ultramafic cumulate like dunite, harzburgite or orthopyroxenite and therefore the observed range of MgO from base to top (from 16 wt% to 1 wt%) is not that dramatic as in the Bushveld-type layered intrusions. But the point here is that the SIC and the Bushveld have distinctly different parental magma compositions. The SIC is produced from quite evolved melt of granodioritic composition whereas the Bushveld from ultramafic composition (so-called B1 melt). Granodioritic composition is simply not able to crystallize ultramafic cumulates such as dunite or orthopyroxenite and this is why they are absent from the SIC (Fig. 1). But this does not mean that the SIC is poorly differentiated. This is just a matter of the original parental melt composition. Instead, the SIC has a rock - quite evolved residual granophyre - which the Bushveld is totally lacking again stressing a difference in their parental magma compositions. So the regular geochemical and mineralogical variations from the felsic norite through the quartz gabbro to granophyre (Fig. 2) should be taken as evidence for the SIC being a well-differentiated body with a quite dramatic degree of its internal fractionation.

Please, note that we have NOT added much about this particular issue in the text since it seems obvious to us and distract attention of readers from major points.

Additional comments to some minor issues are provided below.

Fig. 1. Differences in crystallization sequences and cumulate stratigraphies of the evolved (Sudbury) and primitive (Bushveld-type) parental melts. Note that the evolved one is close to granitic eutectic point and therefore the degree of its differentiation is much more limited compared to the primitive one that is located in the primary stability field of olivine.

Fig. 7. Stratigraphic section, compositional variations and cumulate stratigraphy through the Sudbury Igneous Complex illustrating its well-differentiated nature in terms of crystallization sequence and geochemical indices of magma differentiation (MgO and Mg-number). The Sudbury Intrusive Complex provides the ultimate test for the hypothesis 'no phenocrysts, no post-emplacment differentiation' because it crystallized from a highly superheated and therefore phenocryst-free parental magma, but nonetheless shows remarkable evidence of magmatic differentiation. The existence of the Sudbury Intrusive Complex provides the strongest refutation of the hypothesis: it demonstrates the ability of a phenocryst-free magma to freely differentiate in a crustal magma chamber forming a well-differentiated intrusion showing pronounced phase, modal and cryptic layering (See text for further discussion). Modified after Lightfoot *et al.* (2001). The cumulate stratigraphy is based on petrographic and rock chemical data from Lightfoot *et al.* (2001), Therriault *et al.* (2002) and Zieg & Marsh (2005).

Fig. 2. Latypov, 2009, Journal of Petrology

This is a well written manuscript that addresses a long-standing conundrum concerning the origin of systematic compositional zoning with stratigraphic height in the Sudbury Igneous Complex. Thanks to the efforts of these authors, among others, it is now widely accepted that the SIC magma formed by impact melting about 1.8 billion years ago, but the detailed petrogenesis of the compositional variations have remained rather enigmatic, as nicely documented in the manuscript. **Thanks**

Here the authors present new observations that support an origin by fractional crystallization rather than other types of processes that have been proposed. The key new observation is the physical distribution of blocks of norite that appear to have crystallized on the roof of the SIC and then became dislodged and settled through the magma. Personally I have always been willing to accept a fractional crystallization model for the compositional zoning of the SIC, and these new observations strengthen that interpretation. So the conclusions about the petrologic and magmatic evolution of the SIC seem to be robust, and that part of the paper is well reasoned and clearly presented. **Thanks**

The problem that I have with this paper is the attempt to extrapolate to other planets and wrap the study in a broader significance about early planetary evolution. This aspect of the work is speculative, superficial, and not at all convincing. For example, although the authors provide some citations in line 8, they do not come to grips with the overwhelming lack of measurement-based (i.e., either samples or remote sensing) evidence for large-scale differentiation of lunar impact melt rocks (but see Norman et al 2016 GCA for an alternate example, not cited), or the fundamental physical and thermo-mechanical differences between the crust of the Earth vs. other terrestrial planets. This is especially important here considering the age and unique (for a large terrestrial impact structure) petro-tectonic setting of the SIC. Mars might provide a more apt analogy than the Moon but there is no evidence that I am aware of for strong impact-driven differentiation of Mars beyond possibly the earliest stages of planetary accretion (i.e., formation of the crustal dichotomy by a giant impact).

The reviewer talks as if the SIC is in an unique setting. It is actually nothing special-simply having occurred in a foreland basin.

More specific to the Earth, it appears difficult to physically and chemically isolate only the evolved felsic differentiates of impact melt sheets like those produced in the SIC in a way that would influence the large-scale geochemical evolution of the depleted mantle and continental crust. So while it might be possible that Hadean impact melt sheets were more strongly differentiated than younger examples, the manuscript does not provide adequate support for the speculation that this process may have influenced the large-scale geodynamical evolution of the Earth as implied by lines

122-126.

All what we are saying is that SIC like differentiation will produce felsic "lithologies" and the evidence of the temperature of Hadean zircons is consistent with the T of zircons from the granophyre and quartz gabbro. This does not mean they are inconsistent from formation by endogenic igneous processes. We are explicit that SIC like mechanism would only produce felsic "crust", i.e., proto-continental crust, if the mafic lithologies foundered. We do not claim to be anything but speculative but the same arguments apply to endogenic processes in the Hadean. There are almost no rocks-just zircons from that time! So the same rock evidence satisfies both a exogenic and endogenic process for their origin. We say no more than that.

My recommendation is that this manuscript is better suited to Geology, and that the paper would be strengthened if the discussion maintained its focus on the SIC, perhaps by addressing other unusual aspects such as the economic mineralization, rather than attempting to hype the study beyond what can be reasonably supported in a short-format article.

Some line-by-line comments are provided below.

Line 10: I am not completely convinced that the SIC is the best-preserved impact melt sheet on Earth. It might be the best exposed example, but it has been tectonically and hydrothermally altered. The Chicxulub or even Manicouagan melt sheets might actually be better preserved even if exposure is not as good as the SIC.
Agree, corrected

Line 19 and elsewhere: I am not sure that 'compositional stratigraphy' is the best term to describe the variations seen in the SIC. The term 'stratigraphy' typically implies discrete layers but as shown in Fig. 2, the compositional variation is smooth and systematic such that 'compositional zoning with stratigraphic height' might be a better description.

Sudbury Igneous Complex is a layered intrusion formed from a parental melt of specific composition (granodioritic). Therefore, layered intrusion's terminology can be applied to the SIC as well. A term 'stratigraphy' is commonly used in our field, but you are right that the adjective "compositional" is not good. Better to say "magmatic stratigraphy" or "cumulate stratigraphy".

Line 21 and elsewhere: Personally, I would not describe the compositional variation of the SIC from about 1-7 wt% MgO as 'large-scale differentiation'. Certainly there are some more mafic compositions, but I don't see the evidence for compositionally extreme ultramafic or anorthositic units such as those in layered mafic intrusions such as Bushveld and Stillwater. The SIC seems to be moderately zoned in bulk compositions similar to a lava lake such as Kilauea Iki, rather than strongly differentiated. Please, see a reply above (item 3)

Line 28: replace 'size' with 'diameter' OK

Line 30: With respect to the authors, I just don't think the SIC plays as much role in people's thinking about impact melt sheets as it used to. We now see similar evidence for moderate compositional zoning in the melt sheet at Manicouagan, and the possibility that melt sheets can differentiate is now widely considered as a possibility (as shown by refs cited in the Introduction) even if the scale of that differentiation remains poorly established, especially on other planets with very different crustal compositions and thermal structures. So while the SIC remains a classic locality, I just don't think that the justification for using the SIC as a fundamental planetary analogue is quite as compelling as implied by the authors here. OK

Line 47: 'dramatic' is a bit of an over-statement here. OK

Line 119: As noted above, I would say that large melt sheets 'may' undergo crystal-melt differentiation rather than asserting this as a certainty. Although thermo-mechanical models for efficient differentiation of large melt sheets have been published, I am not aware of any that are differentiated to the extent of layered mafic intrusions such as Bushveld and Stillwater, which is what the authors are implying here. Even the study of an apparently differentiated lunar melt rock by Norman et al 2016 GCA proposed a much more modest process more analogous to filter pressing rather than crystal layering. The trace element variations in the SIC that are documented in Fig. 3 also seems more compatible with a more subdued style of crystal-melt segregation. Please, see a reply above (item 3)

The figures are well drawn and nicely illustrate key observations of the study and relevant aspects of the SIC. Thanks

Reviewer #2 (Remarks to the Author):

A short summary of our changes/replies to the reviewer's comments:

1. Review of potential models: One section starting 'Discussion' is now devoted to analysis of potential models for origin of melanorite bodies discovered in the SIC. Golightly's model about potential incorporation of mafic clasts from basement rocks and the model involving autobrecciation of Sublayers norites are considered. Both models are rejected on the basis that xenolithic mafic clasts and fragments of the Sublayers norites will be denser (2.80-2.85 g/cm³) than the granodioritic impact melt (2.47-2.50 g/cm³) and will therefore sink, rather than float in the melt. (*P.S. We have originally included here a detailed analysis of all six models but the paper started looking as prepared for normal geological journals, not for the Nature Communications. So we had to cut this part and make it shorter but still quite informative*).

2. Shape of melanorite bodies: Yes, this is an important issue. We agree that a term 'angular', which we used was not a good one because real contacts of these bodies with host rocks are almost impossible to see in the field. But what we actually meant to say is that these bodies are 'discrete', as clearly indicated by our mapping. We have not clearly indicated this in the paper. This is enough to discard the many models, which implies that melanorite should form (semi) continuous layers in the stratigraphy of the SIC. This is certainly not the case.

3. Melt parental to melanorite bodies: We did not certainly mean that these bodies were crystallized from some kind of melt of melanoritic composition. Rather, we suggest that they crystallize from granodioritic melt, which has orthopyroxene as a first liquidus phase. Thus, the melanorites are first cumulates produced from this parental granodioritic melt. We have now clearly discussed this in the paper. An issue regarding vaguely symmetrical profile expected for the inward-crystallizing SIC in the frame of our model is also addressed. A simple answer is that we do not see this profile because of the almost complete destruction of the roof sequence.

4. Definition of a term 'melanorite': The term melanorite has been introduced by Lightfoot and Zotov (2005) and Lightfoot (2016) who have found a first body at the Creighton area. It is somewhat misnomer because it is just used to emphasize a higher amount of orthopyroxene in these rocks (30-35%) compared to more normal, felsic norite (20-23%). We cannot think of any better term and therefore their nomenclature is retained here. We have indicated this in the paper.

5. Texture of roof melanorite. Very good point. Fine-grained texture of this rocks is consistent with cooling/quenched of the impact melt against the cold Onaping Formation. This supports our idea this roof melanorite body is located in situ, i.e. where it was formed.

6. Thickness of the SIC. This is where we would prefer to follow Lightfoot (2016) who argues quite convincingly for us that the SIC locally reach a thickness of about 5 km, especially in the South Range which is our major study area. The SIC is mostly 2 km in the North and East Ranges.

7. Implication for the Hadean Earth. This part is now substantially expanded and includes the phase equilibria modelling with two additional figures. The relevance of the SIC to our suggestion regarding making felsic rocks out of impacts from primitive, possibly ultramafic Hadean oceanic crust is now clearly discussed in details. In short, if such a relatively small body as the SIC can differentiate, then it is logical to assume that the same will occur in much larger and more primitive Hadean impact melt sheet. Delamination of dense ultramafic cumulates from these differentiated sheets may play a role in making the Hadean proto-crust increasingly more felsic.

All minor points below are agreed and addressed accordingly in the paper as well.

Overview:

The discovery of enclaves of more melanocratic, more highly-magnesian norite in the upper layers of stratigraphy of the Sudbury Igneous Complex (SIC) represents a useful and important finding in one of the world's most distinctive and important igneous bodies. The explanation for these bodies does indeed pose important questions in terms of the evolution of impact melt sheets, as indicated by the authors. However, I believe that the authors have not done enough here to support their interpretation of in situ internal differentiation having produced melanorites at the roof. Specifically, the analysis of existing models is virtually absent, and the context of how these rocks relate to existing basal melanorite units, and how they would fit into a differentiation model for the rest of the SIC rocks is not really addressed. The nature of the physical evidence is not entirely clear (size and shape of enclaves is important, but precisely how this information was determined is not spelled out), and the manuscript is under-referenced and slightly casual with dimensions and age details. I would recommend revision with emphasis on attention to details such as these to maximize the potential impact of this work.

Detailed comments:

Line Comment

7 The citation superscript (1-8) should be attached to the word 'controversial', rather than the following word, 'whether'.

27, 28 Should read "10s of km", and "100s of km", or better still, "tens of kilometres" and "hundreds of kilometres".

31 Insert comma after "Earth"

33 The age of the SIC should be reported as ca. 1.85 Ga, rather than 1.8; I think 500 million years is worthy of quibbling about. The age also requires a citation, for which the 1984 Krogh et al. source and/or Corfu & Lightfoot (1996) are still suitable; more recent efforts haven't changed anything.

34 The source of the melt sheet temperature data should also be cited; between Grieve, Prevec & Cawthorn, Ivanov, Ariskin, you have a few choices.

50 I think you should avoid "Opx" in the text, and replace with the word orthopyroxene. (In spite of Fig. 2 caption)

51 Similarly, reference to "Creighton, South Range", is overly brief; "in the Creighton area of the SIC South Range" would be more useful.

58 Rather than “no other melanorite bodies occur, as far as 1.5 km etc.”, which reads awkwardly, I’d suggest: “No other melanorite bodies have been found within 1.5 km...”

69 Reference to possible sources of inherited/xenolithic mafic target rocks:

- First, the presence of any inherited target rocks lying around in the melt sheet after thermal equilibration/homogenization would require that the melt sheet was not capable of consuming those (large) fragments, and would contradict the assumption of a superheated melt sheet after those first few seconds/minutes, as it would, by definition, have frozen after trying and failing to consume those xenoliths. It is generally accepted that there were no large target rock fragments left after melt equilibration.
- Second, there are other much better sources of ultramafic rock in the footwall, specifically ultramafic bodies hosted in the North Range footwall both near (Fraser; see Lightfoot) and distal to (Smith et al. 2014) the SIC contact. The East Bull Lake suite are mainly leucogabbroites (see published work by James, Peck & friends), with notoriously little ultramafic component.
- I concur that for these reasons, as well as those you mention (geochemical consistency of your enclaves with other SIC rocks), the enclaves are unlikely to be xenolithic, but you should/can present a much more robust argument for this.

77 The comment that the previously recognized melanorite is 3 km below your reported findings is rather loose with the data; the SIC melt sheet is approximately 2.5 km thick now, preserving lithologies from base to roof, so while there have been suggestions that the melt may have been more extensive/thicker (up to 4.5 km by Ariskin et al., 1999, and Lightfoot (2017) gives the thickness somewhat unhelpfully as 1.5-5 km, depending on variations in floor topography, and his Fig 3.34 addressing this specific topic is not dimensionally quantified, vertically), this is not really relevant to stratigraphic references relating to the observed sequence. A number such as 2 km would still make your point perfectly well, and be consistent with observed stratigraphic constraints.

77-78 A statement such as “Clearly it is physically impossible to relocate such large bodies stratigraphically up-section” is not a useful statement, with no accompanying analysis or citation to sources who have done so. As it is, this comprises your rejection of at least some of the proposed models, with no substantiated critical analysis to accompany it.

For example, the authors have cited Golightly’s suggestion of segregating mafic and felsic components in the post-impact melt phase, a model which superficially would explain bits of mafic material trapped in felsic host, as well as, presumably, bits of felsic material trapped in mafic rocks (which are arguably present in the South Range

as well). I believe vigorous convection within the cooling superheated melt sheet combined with autobrecciation of earlier cumulate material (such as the basal melanorites) is also an option. Neither of these are discussed or ruled out here, which weakens the proposed interpretation. Some of this refutation depends on the available physical data (size & shape of enclaves), which is commented on further below.

88 The reference to the Palisades Sill should be accompanied by a citation to a source (Naslund, for example?).

93 See line 50 comment

97 See line 33 comment

111 500 m of roof melanorite would not represent 10% of the SIC, which has never been proposed as a 5 km thick entity throughout(double its current known thickness); how about "at least 5%", which leaves it 'significant but open to growth'?

285 Insert comma after South Range.

Systemic concerns:

More rigorous analysis of genetic models needed:

As mentioned above (line 77-78), the findings of melanorite in the upper/uppermost SIC melt sheet stratigraphy are potentially explicable by proposed models which have not been refuted in any critical fashion. Specifically:

- Golightly's model would presumably require that the felsic & mafic components were "clast-melt mixtures" (which is probably not consistent with thermal equilibration models in which all clasts should be consumed if prolonged superheat is to be accepted; if it's not, then this is not a problem) which floated to the top and bottom according to gross density of the mixture. In this case, however, the clasts might be expected to be unconsumed target rock lithologies (this is not explicitly stated by Golightly, but seems to be implicit), which you have noted is unlikely (see also comment for line 69).
- Convective brecciation of early roof & floor rocks would be conceivable, with the roof & floor (rather than the hotter interior) being the most likely places to later preserve these trapped blocks. The environment of a superheated melt sheet contained within a steep thermal gradient is as good a place to facilitate vigorous convection as you could ask for in nature, so some density modelling/calculations to illustrate the potential problems with moving very large fragments within the melt would be required in order to convincingly dismiss such a model.

A large part (probably more important to the argument than the geochemistry) of the author's argument for the proximal/overlying origins for these enclaves is their angular shape (which mitigates against an immiscibility origin, for example) and their large size (which mitigates against significant transport distances upwards against gravity); however, you have explicitly stated (lines 136-138) that the fragments are extremely difficult to identify in the field or even petrographically, and that geochemical criteria are the only reliable fingerprints for them. Your argument nonetheless uses their size and shape as critical parameters, even though your sampling density does not allow for the delineation or confirmation of the proposed dimensions of the fragments as shown in Fig. 1. The examples provided in the Extended Data, such as Fig. 2b, showing fist sized fragments of more melanocratic rock, are fine as examples of xenolithic fragments, but not itself convincing as relates to much larger blocks, for which the Fig 2 a photo is not a useful illustration in terms of elucidating lithological contrast. This applies similarly to Fig 1 and 3, where the ability to define the dimensions of the enclaves is not clear. However, the photos do appear to show that you are in fact able to distinguish the darker coloured xenoliths from their surroundings, which provides credibility to your size and shape estimates; this point needs to be made clear, as it appears to contradict the statements cited earlier in this paragraph.

The sample shown in Fig 3 appears to be very rapidly cooled in contrast to those in fig 1 and 2, which look like 'normal' SIC rocks; the sample in Fig 3 is anomalous, and appears to be dyke-like or nearly chilled. This petrographic contrast is noted (line 149) but not discussed.

Consistency with context missing:

The authors may appear to be proposing, then, that the earliest material to have crystallised in the melt sheet was of melanocratic composition. If this is meant to be representative of early liquids, this would be entirely inconsistent with existing models which correlate the bulk composition of the impact target rocks with the compositions of early melt injections into the floor (i.e., Offset dykes) and more directly still with lowermost units of the fallback breccia (Onaping Fm), all of which are quartz dioritic in broad composition, and neither of which are melanocratic nor Mg-rich.

Alternatively, and more likely, the authors are proposing that the SIC cooled and fractionated inwards consistent with the closed system crystallization of a "normal" intrusion or sill, such as the Skaergaard Intrusion (Greenland). This would require an inwards trending, vaguely symmetrical geochemical profile consistent with internal differentiation, which is not obviously the case here (as shown in their Fig 2, for example). I am not suggesting that this case cannot be made on the basis of existing data, but no effort has been made to present such a case here.

In addition, the term melanorite is used very loosely in the manuscript as it relates to existing SIC nomenclature. The basal unit of the melt sheet has been variously named as the Mafic Norite in the North Range, and the Quartz Norite in the South Range, each distinguished by petrographic and compositional peculiarities from each other, and from the overlying so-called Felsic Norite (cf. Lightfoot; various). If we lump the Mafic and Qz Norites together for ease of discussion, this unit is a discontinuous/intermittent basal unit overlying contact sublayer embayments, which comprise basal breccias. The implication is that the more melanocratic norite overlying the embayments is then locally-formed, or perhaps residual opx-cumulate norite which has been eroded away where not protected by the embayment mouth structures. It appears to be cut/intruded by the underlying sublayer, which would be consistent with any of the proposed models. It is not made at all clear how the authors are defining their term melanorite in this context.

Finally, the last paragraph of the manuscript (lines 113-128) are devoted to the potential significance of these findings as they relate to Hadean crustal evolution, specifically the concept that early impact melts could be generating felsic crust. This discussion does not qualify how the proposed model of internal differentiation would differ from any other model proposed for Sudbury which also involve the production of differentiated mafic and felsic melts/rocks from an impact process (i.e., the process may differ, but the outcomes are the same; how does this help us in the Hadean? This needs to be made clear.). Since the SIC melt was a product of homogenization of already very felsic (largely granitic, s.l.) target rocks, it's relevance to making felsic rocks out of impacts into primitive, possibly ultramafic Hadean oceanic crust, is also unclear.

Reviewer #3 (Remarks to the Author):

Replies to all comments are inserted into the text below.

The paper's thesis, that impact melt sheets were an important geologic product on the early Earth and that differentiation of them was likely and, thus, important in the evolution of Earth's crust, is a finding that should shape any geologist's evaluation of early Earth processes. For that reason, the paper is a compelling contribution of broad interest. I append comments that are mostly minor and intended to help the authors polish the submitted paper. The most important comments are (9), (10), and (12). I suspect those comments can be addressed without much work and that the paper can be published soon thereafter.

1)Opening paragraph. The SIC is described as the "best preserved impact melt sheet on Earth." The Chicxulub melt sheet may be better preserved, although it is buried. The SIC is dramatic because it is "exposed" and "accessible." Minor rewording may be appropriate to better convey the status of SIC. **Ok, done**

2)Opening paragraph. For this journal's audience, it will be important to define "melanorite." **Ok, done, but in the text**

3)Opening paragraph. The description of Hadean impact melt sheets and the implied effect on Earth's crust should probably cite a paper by Grieve, 1980, in Precambrian Research. **Ok, included**

4)Opening paragraph. Here or elsewhere, the paper may need to refer to other work to convince readers that impact melt sheets may be an abundant geologic product of the Hadean. A relatively recent example was published by Abramov and others, 2013, in Chemie der Erde. That paper or the references therein may be suitable. There was another recent paper by Marchi and others that appeared in Nature, Science, or one of their recent offspring. **Ok, included**

5)Second paragraph. The opening line is important. The geologic process of impact cratering was a major process affecting Hadean Earth. It is a process ignored by most investigators of the Hadean. The current manuscript may fundamentally affect the way geologists view the early Earth. **Thanks**

6)Line 31: The SIC is dramatic because it is "exposed" and "accessible." Minor rewording may be appropriate to better convey the status of SIC. **Ok, corrected**

7)Lines 34 through 36. Are the commonly-used lithological names (norite, granophyre, and quartz gabbro) strictly correct? If not, it may be better to provide readers with both the historical terminology and petrologically-correct terminology,

so that readers are not compositionally misled. The terms are fine, well-accepted for the SIC.

8)Line 47: The word “dramatic” may be inappropriate. Ok, deleted.

9)Lines 49-51: The observed properties (medium-grained, sulphide-bearing melnocratic norite) is a clue that prompts a question. Is the sulphide consistent with the model proposed in this paper? Sudbury sulphide has two general sources: (a) it settles to the bottom of the melt sheet due to differentiation and (b) is produced hydrothermally during the solidification and cooling of the melt sheet. In the case of (a), one wonders how it could be trapped in a type of rock that formed at the top of the melt sheet (lines 81-82) and/or in rock that sometimes still occurs near the top of the melt sheet (Figure 1d). In the case of (b), why would the sulphide preferentially impregnate the melanorite blocks? An explanation of the sulphide is needed that is consistent with the paper’s model.

We have now touched this issue now in the text. Blebs of sulphides dispersed in melanorites apparently indicate that the impact melt has reached sulphide liquid immiscibility before onset of its crystallization. This is also compatible with the fact that there are sulphides in the chilled upper rocks of the SIC (formerly Basal Onaping) and in both phases of the Offset dikes.

10)Line 67: The description of “angular blocks” would be compelling evidence of the model proposed in the paper. Yet, evidence of “angular” morphology seems weak. In the “Methods” section (lines 136-140), readers are told “visual identification of the melanorite bodies in the field is extremely difficult.....Geochemistry appears to be the only tool that allows the confident identification and mapping of melanorite bodies in the SIC.” If they are not confidently mapped in the field, how confident is the assessment of an “angular” morphology? Can the authors clarify?

Yes, this is an important issue. We agree that a term ‘angular’, which we used was not a good one because real contacts of these bodies with host rocks are almost impossible to see in the field. But what we actually meant to say is that these bodies are ‘discrete’, as clearly indicated by our mapping. We have not clearly indicated this in the paper. This is enough to discard the Models 4, 5 and 6, which implies that melanorite should form (semi) continuous layers in the stratigraphy of the SIC. This is certainly not the case.

11)Lines 69-70: The isotopic and geochemical evidence of an SIC origin of the blocks is compelling. Ok

12)Lines 81-82: Did the SIC have sloping walls? If so, could the melanorite in the clasts have solidified high up on the basin walls (which is still effectively the base of the melt sheet and, thus, likely to contain sulphide) and then slumped into the melt sheet? All current theoretical models indicate that the Sudbury impact melt sheet was nearly-horizontal body with no steeply-dipping sidewalls. So no problems from this side.

13)Line 85: What evidence exists for SIC cooling via circulating ocean water? While the papers cited may describe models for the Onaping that involve melt-seawater interaction, I do not recall the papers demonstrating (a) that seawater was the principal cooling mechanism of the underlying SIC or that (b) the sea was the dominant source of water flowing around and through SIC. The authors do not need to (a) and (b) to make their case. Thus, the simple solution is to simply delete the phrase “cold circulating ocean water.” It is sufficient to say that the top of the melt sheet (just like a lava lake or lava flow) cooled and solidified before the interior of the melt sheet. **Three papers below to which we refer in the text are suggesting the SIC cooling via circulating ocean water.**

Grieve, R. A. F., Ames D. E., Morgan, J. V. & Artemieva, N. The evolution of the Onaping Formation at the Sudbury impact structure. *Meteoritics & Planetary Science* 45, 759–782 (2010)

Anders, D., Osinski, G. R., Grieve, R. A. F. & Brillinger, D. T. M. The Basal Onaping intrusion in the North Range of the Sudbury impact structure: roof rocks of the Sudbury Igneous Complex: *Meteoritics & Planetary Science* 50, 1577–1594 (2015).

Ubide, T., Guyett, P. C., Kenny, G. G., O'Sullivan, E. M., Ames, D. E., Petrus, J. A., Riggs, N. & Kamber, B. S. Protracted volcanism after large impacts: evidence from the Sudbury impact basin. *Journal of Geophysical Research* 122, 701–728 (2017).

14)Lines 117-120. The Vaughn and Head reference (PSS 2014) is fine. The Hurwitz and Kring reference (EPSL 2015) is not appropriate. I suspect the authors are confusing that paper with another (Hurwitz and Kring, JGR 2014). **Yes, corrected.**

15)Same lines. “South Pole Aitken” should be written “South Pole-Aitken.” **Ok, corrected**

16)Same lines. This section is another good place for the Grieve (1980) reference described in note (3) above. **Ok, added**

17)Same lines. The text overlooks the potential consequences of bulk melt composition. Does the ease with which differentiation occurs depend on melt composition? How does the Sudbury melt composition compare with those inferred for the Hadean or on other planetary bodies. Also, the text overlooks the potential consequences of planetary gravity. If I recall correctly, Paul Warren argued the propensity for differentiation depends on gravity, so melt sheet differentiation that occurs on the Earth may not necessarily occur in a melt sheet of the same size on another planetary body. This point is not central to the paper’s argument, but on how the paper’s result can be used elsewhere. It is, thus, a minor point that the authors can address with simple rewording. **Good points, we have addressed them it in the last section dealing with the Moon.**

18) Lines 126-128. The text seems to contradict a finding by Wielicki and others (EPSL 2012). The text refers to a paper by Kenny and others (Geology 2016). If the Geology paper demonstrated the older EPSL work was flawed, then the content of lines 126-128 is fine. If, however, a discrepancy between the Geology and EPSL papers remains, the authors should refer to both works. **Yes, Kenny and others (Geology 2016) show that the older EPSL work was flawed. Wielicki et al only looked at zircons from the norite. Kenny included zircons from the granophyre, which have the required low temperatures and overlap those of Hadean zircons.**

19) Figure 4a. The morphology of the crater cross-section is probably incorrect. A central peak complex crater is illustrated. If Sudbury is as large as Chicxulub (or larger), then it had a peak-ring basin morphology like the well-exposed Schrodinger basin on the Moon. Grieve and others (JGR 1991) may have produced the first illustration of a reconstructed Sudbury, although there are other, more recent examples of peak-ring basin cross-sections. **Sorry, we are not following this. Everything seems to be correct to us.**

Reviewer #1 (Remarks to the Author):

I am satisfied with the revisions and responses to the previous reviews, and recommended publication of the manuscript with no further changes.

Reviewer #2 (Remarks to the Author):

I am satisfied that the authors have addressed my concerns relating to the geological context with the framework of SIC geology appropriately and diligently, which will hopefully result in a more robust impact on the Sudbury and international geological community as a consequence. I have identified only a handful of minor editorial changes, listed below, but I am otherwise satisfied with the revised manuscript.

Line 168: should be regarded as an overlooked enigma; insert "an" ahead of the word overlooked

Line 169: I suggest, in line with comments from other reviewers, that the word "facilitated" be inserted ahead of the word "by"; this way, while ocean water can help with the cooling, the presence or absence of ocean water is not critical to the argument.

Line 539: I suspect that the last name listed in the first line of acknowledgements may be R. Keays (Reid), in which case it is misspelled as R. Keys, which he will not like. If it's someone else, then never mind.